# Reduction-based Pseudo-label Generation for Instance-dependent Partial Label Learning

**Congyu Qiao**[1,2], **Ning Xu**[1,2,*], **Yihao Hu**[1,2], and **Xin Geng**[1,2]

[1] School of Computer Science and Engineering, Southeast University, Nanjing 210096, China
[2] Key Laboratory of New Generation Artificial Intelligence Technology and
Its Interdisciplinary Applications (Southeast University), Ministry of Education, China
`{qiaocy, xning, yhhu, xgeng}@seu.edu.cn`

## Abstract

*Instance-dependent Partial Label Learning* (ID-PLL) aims to learn a multi-class predictive model given training instances annotated with candidate labels related to features, among which correct labels are hidden fixed but unknown. The previous works involve leveraging the identification capability of the training model itself to iteratively refine supervision information. However, these methods overlook a critical aspect of ID-PLL: within the original label space, the model may fail to distinguish some incorrect candidate labels that are strongly correlated with features from correct labels. This leads to poor-quality supervision signals and creates a bottleneck in the training process. In this paper, we propose to leverage reduction-based pseudo-labels to alleviate the influence of incorrect candidate labels and train our predictive model to overcome this bottleneck. Specifically, reduction-based pseudo-labels are generated by performing weighted aggregation on the outputs of a multi-branch auxiliary model, with each branch trained in a label subspace that excludes certain labels. This approach ensures that each branch explicitly avoids the disturbance of the excluded labels, allowing the pseudo-labels provided for instances troubled by these excluded labels to benefit from the unaffected branches. Theoretically, we demonstrate that reduction-based pseudo-labels exhibit greater consistency with the Bayes optimal classifier compared to pseudo-labels directly generated from the training predictive model.

## 1 Introduction

Partial Label Learning (PLL), a typical weakly supervised learning paradigm, aims to build a predictive model that assigns the correct label to unseen instances by learning from training instances annotated with a candidate label set that obscures the exact correct label [5, 3, 49]. The necessity to learn from such weak supervision naturally arises in ecoinformatics [21, 32], web mining [22], multimedia content analysis [50, 2], and other domains, owing to the difficulty in collecting large-scale high-quality datasets.

PLL has been studied along two different routes: the identification-based route[13, 26, 21, 3, 49, 24, 7, 51, 35, 37], which treats correct labels as the latent variable and tries to identifies them, and the average-based route [12, 5, 52, 23], which treats all candidate labels equally. To facilitate practical PLL algorithms, some researchers have focused on instance-dependent PLL (ID-PLL), where incorrect labels related to features are likely to be selected as candidate labels, and tackled this challenge by following the identification-based route. [44] explicitly estimate the label distribution aligned with the model output on candidate labels through variational inference. [39] induce a

---

[*]Corresponding author.

contrastive learning framework from ambiguity to refine the representation of the model. [28] model the generation process of instance-dependent candidate labels by leveraging prior knowledge in the model output [42] propose a theoretically-guaranteed method that progressively identifies incorrect candidate labels by leveraging the margin between the model output values on candidate labels.

These previous approaches involve leveraging the identification capability of the training model itself to iteratively refine supervision information. They exercise this capability through the outputs [44, 28, 11, 38] or representations [39] from the training model. However, these methods overlook a critical issue in ID-PLL: some incorret candidate labels strongly related to features may not be distinguished with correct labels by the model within the original label spaces. As a result, the model generates increasing amounts of incorrect identification information, further leading to the training bottleneck. Unlike previous approaches that iteratively train the predictive model under the influence of misleading supervision before performing identification, we prioritize eliminating the influence of misleading candidate labels within label subspaces, and then train an auxiliary model free from this influence to provide accurate identification information.

In this paper, we propose to utilize reduction-based pseudo-labels to mitigate the influence of incorrect labels and train our predictive model. Specifically, reduction-based pseudo-labels are generated by aggregating the outputs of a multi-branch auxiliary model, with each branch trained in a label subspace that excludes certain labels. This approach allows each branch to explicitly avoid the interference of the excluded labels. Consequently, instances affected by these excluded labels can benefit from the pseudo-labels provided by the corresponding branch. Note that the auxiliary model will not be involved in the testing time, which is similar to [44, 28]. Moreover, given mild assumptions, we demonstrate that pseudo-labels generated from the model trained in the label subspace could exhibit greater consistency with the Bayes optimal classifier compared to those from the predictive model itself. Our contributions can be summarized as follows:

- Theoretically, we prove that reduction-based pseudo-labels generated from the model trained in label subspace could be more consistent with the Bayes optimal classifier than that from the predictive model itself.
- Practically, we propose a novel pseudo-label generation approach named RPLG to deal with the ID-PLL problem, which utilizes the multi-branch auxiliary model with each branch trained in a label subspace to alleviate the impact of incorrect candidate labels strongly related to the instances and disturbing the training process.

## 2 Related Work

PLL has been studied along two different routes: the identification-based route and the average-based route. Identification-based approaches [13, 26, 21, 3, 49] are intuitional since they aim to gradually identify latent correct labels from candidate labels or eliminate incorrect labels out of candidate labels, which is also commonly referred to as disambiguating. In recent years, most researchers have attempted to tackle the problem of PLL along the identification-based route and achieved tremendous improvements. Average-based approaches [12, 5, 52] tend to deal with a learning objective where all candidate labels are treated equally. A typical average-based approach is based on distance between instances and predict the label of a unseen instance by voting among the candidate labels of its nearest neighbors in the feature space. Very recently, [23] theoretically and empirically demonstrate their proposed approach along the average-based route have an advantage in robustness. In this paper, we still choose to follow the route of identification-based routes to handle ID-PLL.

Identification in PLL are performed by various means. In traditional PLL, to which deep neural networks (DNN) have not been applied, [26] maximize the margin between the maximum modeling output from candidate labels and that from non-candidate labels to implicitly identify correct labels, while [49] further try to directly maximize the margin between the correct label and all labels. [21] propose the Logistic Stick-Breaking Conditional Multinomial Model to maximum the likelihood of candidate label, marginalizing away the latent correct labels. [53, 6, 34, 43] leverage the topological information in the feature space to iteratively update the confidence of each candidate label or the label distribution, from which we could determine correct and incorrect labels. In deep PLL, [46] enhance the identification ability of the model by designing an entropy-based regularization term and temporally assembling predictions of the model as the guidance of train-

ing. [24] propose a progressive identification method that normalizes the output of the model on candidate labels in each epoch as the weight in the cross-entropy loss. [47] reduce identification error by introducing a network-cooperation mechanism. [7] build a risk-consistent estimator and a classifier-consistent estimator relying on the process of identidication. [36] propose a loss function with a parameter weighting losses on candidate and non-candidate labels, implicitly enhancing idetification. [51, 10, 25] improve identification through class activation value, semantic label representation and structured representation provided by deep graph neural networks, respectively. [35] propose a contrastive learning framework that could adapt to PLL and leverage the class prototypes in the framework for identification. [37] augment the data to obtain conformed label distributions capable of identification to perform manifold regularization. However, these works have been actually considering PLL by unrealistically assuming that incorrect candidate labels are uniformly sampled while the phenomenon of instance dependence is widely observed in various fields [40, 44, 15] including PLL.

From [44], identification in ID-PLL, which is more practical, begins to be noticed. [44] explicitly estimate the label distribution [8, 45, 16, 14] for each instance, which reflect possibility that a label is selected into candidate sets through variational inference. [39] induce a contrastive learning framework from ambiguity and obtain identifiable representation in ID-PLL. [28] model the generation process of instance-dependent candidate labels to perform maximum-a-posterior, implicitly identifying the latent correct labels with prior information. [42] propose a theoretically-guaranteed method leveraging the margin between the model output values on candidate labels to progressively identify incorrect candidate labels. [11] perform identification with selection of well-disambiguated samples. [38] build a self-distillation framework rectifying the identification results of the teacher model to enhance the reliability of distilled knowledge. In this paper, we propose to explicitly alleviate the influence of some incorrect candidate labels hard to be distinguished on the models through pseudo-labels generated in specified label subspace, further enhancing the identification process.

## 3 Proposed Method

### 3.1 Preliminaries

First of all, we briefly introduce some necessary notations. Let $\mathcal{X} = \mathbb{R}^q$ be the $q$-dimensional instance space and $\mathcal{Y} = \{1, 2, ..., c\}$ be the label space with $c$ class labels. Given the PLL training dataset $\mathcal{D} = \{(\boldsymbol{x}_i, S_i) | 1 \leq i \leq n\}$ where $\boldsymbol{x}_i \in \mathcal{X}$ denotes the $q$-dimensional instance and $S_i \subset \mathcal{Y}$ denotes the candidate label set associated with $\boldsymbol{x}_i$. In PLL, the correct label $y_{\boldsymbol{x}_i}$ of the instance $\boldsymbol{x}_i$ must be in the candidate label set, i.e., $y_{\boldsymbol{x}_i} \in S_i$, and each candidate label set $S_i$ should not be the empty set nor the whole label set, i.e., $S_i \notin \{\emptyset, \mathcal{Y}\}$. Besides, we do not consider the case that the candidate label set $S_i$ only has the correct label $y_{\boldsymbol{x}_i}$ in this paper, namely, $S_i \neq \{y_{\boldsymbol{x}_i}\}$, For each candidate label set $S_i$ in the training dataset, we also use the logical label vector $\boldsymbol{l}_i = [l_i^1, l_i^2, ..., l_i^c]^\top \in \{0, 1\}^c$ to represent whether the label $j$ is one of the annotated labels, i.e., $l_i^j = 1$ if $j \in S_i$, otherwise $l_i^j = 0$.

Let the posterior probability vector $\eta(\boldsymbol{x}) = [\eta^1(\boldsymbol{x}), \eta^2(\boldsymbol{x}), \ldots, \eta^c(\boldsymbol{x})]$ with $\eta^j = p(y = j | \boldsymbol{x})$ denoting the posterior probability of the label $j$ given the instance $\boldsymbol{x}$. A Bayes optimal classifier $\eta^\star$ can be calculated using $\eta$, i.e., $\eta^\star(\boldsymbol{x}) = \arg \max_{j \in \mathcal{Y}} \eta^j(\boldsymbol{x})$.

Moreover, let $\tilde{\mathcal{Y}} \subseteq \mathcal{Y}$ be the labels excluded from the label space $\mathcal{Y}$ and $\eta'(\boldsymbol{x}) = [\eta'^1(\boldsymbol{x}), \eta'^2(\boldsymbol{x}), \ldots, \eta'^c(\boldsymbol{x})]$ be the posterior probability in a label subspace without $\tilde{\mathcal{Y}}$. The $j$-th element of $\eta'(\boldsymbol{x})$ denotes the posterior probability of the label $j$ given the instance $\boldsymbol{x}$ and $y \notin \tilde{\mathcal{Y}}$, i.e.,

$$
\eta'^j(\boldsymbol{x}) = p(y = j | \boldsymbol{x}, y \notin \tilde{\mathcal{Y}}) = \begin{cases} \dfrac{\eta^j(\boldsymbol{x})}{1 - \sum_{k \in \tilde{\mathcal{Y}}} \eta^k(\boldsymbol{x})}, & \text{if } j \notin \tilde{\mathcal{Y}} \\ 0, & \text{otherwise.} \end{cases}
\tag{1}
$$

We consider the task of PLL is to learn a score function $f : \mathcal{X} \mapsto \Delta^{c-1}$, where $\Delta^{c-1}$ denotes the $c$-dimension simplex, as our classifier with its prediction $h(\boldsymbol{x}) = \arg \max_{j \in \mathcal{Y}} f_j(\boldsymbol{x})$ consistent with that of the Bayes optimal classifier $\eta^\star(\boldsymbol{x})$. During the training process, the classifier $f$ could be

considered to be optimized by the following objective:

$$\mathcal{L}(f(\mathbf{X}; \boldsymbol{\Theta}), \mathbf{Q}) = -\frac{1}{n} \sum_{i=1}^{n} \ell(f^j(\boldsymbol{x}_i; \boldsymbol{\Theta}), \boldsymbol{q}_i), \tag{2}$$

where the classifier $f$ is parameterized by $\boldsymbol{\Theta}$, $\ell$ denotes the cross-entropy function, $\mathbf{Q} = [\boldsymbol{q}_1, \boldsymbol{q}_2, \ldots, \boldsymbol{q}_n]^\top$ is the pseudo-label matrix with each element $\boldsymbol{q}_i = [q_i^1, q_i^2, \ldots, q_i^c]$ denoting the pseudo-label of the instance $\boldsymbol{x}_i$, satisfying $\sum_{j=1}^c q_i^j = 1$ and $q_i^j = 0$ if $j \notin S_i$. Since our target is to make the prediction of the classifier $f$ consistent with that of the Bayes optimal classifier $\eta^\star$, the pseudo-label $\boldsymbol{q}_i$ should put the most mass on the label predicted by the Bayes optimal classifier $\eta^\star$, i.e., $\arg\max_{j \in \mathcal{Y}} q_i^j = \eta^\star(\boldsymbol{x})$, which is a very challenging task under instance-dependent PLL.

## 3.2 Overview

To begin with, we provide a formal definition of disturbing incorrect labels, which are labels that the predictive model finds challenging to identify as incorrect. We prove that a model trained in a label subspace excluding these disturbing incorrect labels can produce pseudo-labels more consistent with the Bayes optimal classifier for instances whose candidate label sets include these disturbing incorrect labels. Moreover, we establish a boundary for the conditional probability that the pseudo-labels of these instances are consistent with the Bayes optimal classifier.

Motivated by these theoretical results, we propose RPLG, which leverages pseudo-labels generated in label subspaces, i.e., reduction-based pseudo-labels, to train our predictive model. Reduction-based pseudo-labels are derived from the outputs of an auxiliary multi-branch model. Each branch is trained within a distinct label subspace that explicitly excludes specific labels. To generate reduction-based pseudo-labels, we employ a meta-learned weight vector to fuse the outputs of all branches.

## 3.3 The RPLG Approach

We first introduce the disturbing incorrect labels, which are hard to be distinguished as incorrect labels according to the output of the predictive model during training. These labels pose great challenge to generate pseudo-labels $\boldsymbol{q}_i$ consistent with the Bayes optimal classifier $\eta^\star(\boldsymbol{x}_i)$.

**Definition 1.** *(($\tau, f, \epsilon$)-disturbing incorrect label) An incorrect label $j$ is said to be ($\tau, f, \epsilon$)-disturbing for the predictor $f$ on some instance $\boldsymbol{x}$ with $\eta^\star(\boldsymbol{x}) \neq j$ if $\exists \epsilon \in (0, 1)$, $\forall j \in \mathcal{Y}$, $\max_{\boldsymbol{x}} |f^j(\boldsymbol{x}) - \eta^j(\boldsymbol{x})| \leq \epsilon$ and $\exists \tau \in (0, \min\{1, 2\epsilon\}]$, the posterior $\eta^{\eta^\star(\boldsymbol{x})}(\boldsymbol{x}) - \eta^j(\boldsymbol{x}) \leq \tau$.*

Here, $\tau$ indicates the degree that the posterior $\eta^j(\boldsymbol{x})$ approaches $\eta^{\eta^\star(\boldsymbol{x})}(\boldsymbol{x})$. The smaller its value, the easier the label $j$ is selected into the candidate label set. $\epsilon$ indicates the degree that the predictive model $f$ approximates the posterior $\eta$. According to [42], if $\eta^{\eta^\star(\boldsymbol{x})}(\boldsymbol{x}) - \eta^j(\boldsymbol{x}) > 2\epsilon$, we could distinguish label $j$ as incorrect labels according to the output of the predictive model. Naturally, a problem arises: how should we handle those samples with disturbing incorrect candidate labels which satisfiy $\eta^{\eta^\star(\boldsymbol{x})}(\boldsymbol{x}) - \eta^j(\boldsymbol{x}) < 2\epsilon$.

Then we start with analyzing the pseudo-labels of the instances, whose disturbing labels is denoted by $\tilde{\mathcal{Y}}$. An auxiliary model $\varphi$ is considered to train without the influence of disturbing labels. On mild assumptions, we prove that the pseudo-label provided by the auxiliary model $\varphi$ in a specific label subspace has more chance to be consistent with the Bayes optimal classifier than the predictive model. The proof can be found in Appendix A.1.

**Theorem 1.** *Let $\mathcal{J}(\tilde{\mathcal{Y}}) = \{\boldsymbol{x} \mid \forall j \in \tilde{\mathcal{Y}}, j \text{ is a } (\tau, f, \epsilon)\text{-disturbing incorrect label for } \boldsymbol{x} \text{ with } y \neq j, \text{ and } \forall j \notin \tilde{\mathcal{Y}}, j \text{ is not a } (\tau, f, \epsilon)\text{-disturbing incorrect label for } \boldsymbol{x}\}$. Suppose that a model $\varphi$ trained without the label space $\tilde{\mathcal{Y}}$ satisfies $\exists \epsilon' \in (0, \min\{1, \min_{\boldsymbol{x} \in \mathcal{J}(\tilde{\mathcal{Y}})} \frac{(\eta^{\eta^\star(\boldsymbol{x})}(\boldsymbol{x}) - \eta^b(\boldsymbol{x}))(\eta^{\eta^\star(\boldsymbol{x})}(\boldsymbol{x}) - \eta^a(\boldsymbol{x}))}{4\epsilon(1 - \sum_{j \in \tilde{\mathcal{Y}}} \eta^j(\boldsymbol{x}))}\})$ with $a = \arg\max_{j \in \tilde{\mathcal{Y}}} \eta^j(\boldsymbol{x})$ and $b = \arg\max_{j \notin \{y\} \cup \tilde{\mathcal{Y}}} \eta^j(\boldsymbol{x})$, $|\varphi^j(\boldsymbol{x}) - \eta'^j(\boldsymbol{x})| \leq \epsilon'$, we could obtain:*

$$p(\eta^\star(\boldsymbol{x}) = \arg\max_{j \in \mathcal{Y}} q^j | \boldsymbol{x} \in \mathcal{J}(\tilde{\mathcal{Y}})) \leq p(\eta^\star(\boldsymbol{x}) = \arg\max_{j \in \mathcal{Y}} q'^j | \boldsymbol{x} \in \mathcal{J}(\tilde{\mathcal{Y}})). \tag{3}$$

Theorem 1 inspires us that we can decouple the training of the model from the generation model of pseudo-labels. By introducing some auxiliary model trained in the absence of the label subspace $\tilde{\mathcal{Y}}$,

we can generate pseudo-labels with better Bayesian consistency on the samples in $\mathcal{J}(\tilde{\mathcal{Y}})$. Here, $\tilde{\mathcal{Y}}$ is not a point-wise concept. In fact, given $\tilde{\mathcal{Y}}$, we could obtain a set consisting of instances whose $(\tau, f, \epsilon)$-disturbing incorrect labels exactly constitute the label set $\tilde{\mathcal{Y}}$.

Additionally, we further analyze the chance that the pseudo-label provided by the auxiliary model $\varphi$ is consistent with the Bayes optimal classifier. We assume the Tsybakov condition [33, 54] holds around the margin of the decision boundary of the true posterior in the multi-class scenario.

**Assumption 1.** *(multi-class Tsybakov condition)* $\exists C, \lambda > 0$ *and* $\exists t_0 \in (0, 1]$*, such that for all* $t \leq t_0$*,*

$$p(\eta^{\eta^{\star}(\boldsymbol{x})}(\boldsymbol{x}) - \eta^{s(\boldsymbol{x})}(\boldsymbol{x}) \leq t) \leq Ct^{\lambda}, \tag{4}$$

*where* $s(\boldsymbol{x}) = \arg\max_{j \in \mathcal{Y}, j \neq \eta^{\star}(\boldsymbol{x})} \eta^j(\boldsymbol{x})$ *denotes the second best prediction of* $\eta(\boldsymbol{x})$*.*

Under Assumption 1, we could prove the pseudo-label provided by the auxiliary model $\varphi$ has a good chance to be consistent with the Bayes optimal classifier. The proof can be found in Appendix A.2.

**Theorem 2.** *Suppose that for* $\boldsymbol{x} \in \mathcal{J}(\tilde{\mathcal{Y}})$*, its posterior* $\eta(\boldsymbol{x})$ *fulfills Assumption 1 for constants* $C, \lambda > 0$ *and* $t_0 \in (0, 1]$*. Suppose that a model* $\varphi$ *trained without the label space* $\tilde{\mathcal{Y}}$ *satisfies* $\exists \epsilon' \in (0, \min\{1, \min_{\boldsymbol{x} \in \mathcal{J}(\tilde{\mathcal{Y}})} \frac{(\eta^{\eta^{\star}(\boldsymbol{x})}(\boldsymbol{x}) - \eta^b(\boldsymbol{x}))(\eta^{\eta^{\star}(\boldsymbol{x})}(\boldsymbol{x}) - \eta^a(\boldsymbol{x}))}{4\epsilon(1 - \sum_{j \in \tilde{\mathcal{Y}}} \eta^j(\boldsymbol{x}))}\})$ *with* $a = \arg\max_{j \in \tilde{\mathcal{Y}}} \eta^j(\boldsymbol{x})$ *and* $b = \arg\max_{j \notin \{y\} \cup \tilde{\mathcal{Y}}} \eta^j(\boldsymbol{x})$*,* $|\varphi^j(\boldsymbol{x}) - \eta'^j(\boldsymbol{x})| \leq \epsilon'$*, we could obtain:*

$$p(\eta^{\star}(\boldsymbol{x}) = \arg\max_{j \in \mathcal{Y}} q'^j | \boldsymbol{x} \in \mathcal{J}(\tilde{\mathcal{Y}})) \geq 1 - C[O(\epsilon\epsilon')]^{\lambda} \tag{5}$$

Our theoretical insight inspires a new algorithm for the generation of the pseudo-label $\boldsymbol{q}_i$ in the optimization objective Eq. (2). To begin with, we decompose the pseudo-label $\boldsymbol{q}_i$ into the basic pseudo-label $\boldsymbol{\mu}_i$ and the reduction-based pseudo-label $\boldsymbol{v}_i$, i.e.,

$$\boldsymbol{q}_i = \alpha\boldsymbol{\mu}_i + (1 - \alpha)\boldsymbol{v}_i, \tag{6}$$

where $\alpha$ is a trade-off hyper-parameter to decide the influence of the introduced reduction-based pseudo-labels. The basic pseudo-label $\boldsymbol{\mu}_i$ is initialized with uniform weights on candidate labels and then could be calculated by using the outputs of the predictive model $f$:

$$\mu_i^j = \begin{cases} \dfrac{f^j(\boldsymbol{x}_i; \boldsymbol{\Theta})}{\sum_{k \in S_i} f^k(\boldsymbol{x}_i; \boldsymbol{\Theta})} & \text{if } j \in S_i, \\ 0, & \text{otherwise,} \end{cases} \tag{7}$$

which puts more weights on more possible candidate labels [24]. The reduction-based pseudo-label $\boldsymbol{v}_i$ can be obtained by the output of the model trained without the influence of the label from 1 to $j$ formulated as $\mathbf{U}_i$ and a vector $\boldsymbol{w}_i$ to weight these output:

$$\boldsymbol{v}_i = \boldsymbol{w}_i \mathbf{U}_i. \tag{8}$$

Here, $\mathbf{U}_i = [\boldsymbol{u}_i^1, \boldsymbol{u}_i^2, \dots, \boldsymbol{u}_i^c]^{\top} \in \mathbb{R}^{c \times c}$ is a reduction-based matrix, which is an intermediate variable combined with the vector $\boldsymbol{\omega}_i$ to generate the reduction-based pseudo-label $\boldsymbol{v}_i$ in Eq. (8). The $j$-th row of $\mathbf{U}_i$ is initialized with uniform weights on candidate labels without label $j$ and then calculated by:

$$u_i^{j,r} = \begin{cases} \dfrac{\varphi^r(\boldsymbol{z}_i; \boldsymbol{\Omega}_j)}{\sum_{k \in S_i \setminus \{j\}} \varphi^k(\boldsymbol{z}_i; \boldsymbol{\Omega}_j)} & \text{if } r \in S_i \setminus \{j\}, \\ 0, & \text{otherwise,} \end{cases} \tag{9}$$

where $\varphi$ is an auxiliary model parameterized by $\{\boldsymbol{\Omega}_j\}_{j=1}^c$ to form $c$ branches, and $\boldsymbol{z}_i$ is the extracted features given the instance $\boldsymbol{x}_i$. The $j$-th branch $\varphi(\cdot; \boldsymbol{\Omega}_j)$ is trained without $j$ in the label space $\mathcal{Y}$, and the loss function for the auxiliary model $\varphi$ could be formulated by:

$$\mathcal{L}^{\text{aux}}(\{\varphi(\boldsymbol{Z}; \boldsymbol{\Omega}_j)\}_{j=1}^c, \{\mathbf{U}_i\}_{i=1}^n) = -\frac{1}{n} \sum_{i=1}^n \sum_{j=1}^c \ell(\varphi(\boldsymbol{z}_i; \boldsymbol{\Omega}_j), \boldsymbol{u}_i^j). \tag{10}$$

The $j$-th branch uses its previous outputs normalized on candidate labels without $j$ as its supervision in the next epoch, which is similar to the initialization and training mode in PRODEN [24]. In this

Table 1: Classification accuracy (mean±std) of each comparing approach on benchmark datasets for instance-dependent PLL.

| Dataset | FMNIST | KMNIST | CIFAR10 | CIFAR100 | TinyImageNet |
|---|---|---|---|---|---|
| RPLG | **91.41±0.13%** | **96.85±0.11%** | **87.53±0.21%** | **65.03±0.21%** | **40.74±0.64%** |
| DIRK | 91.02±0.23%• | 96.21±0.49%• | 84.63±0.22%• | 58.17±0.20%• | 25.77±1.55%• |
| SDCT | 90.98±0.45%• | 96.01±0.34%• | 86.50±0.13%• | 60.95±0.35%• | 36.50±0.35%• |
| POP | 90.12±0.35%• | 95.03±0.83%• | 86.23±0.36%• | 60.71±0.16%• | 39.27±0.86%• |
| IDGP | 90.87±0.41%• | 95.98±0.51%• | 86.43±0.23%• | 64.38±0.27%• | 32.21±1.14%• |
| ABLE | 90.43±0.09%• | 95.37±0.07%• | 85.11±0.24%• | 61.21±0.37%• | 23.60±0.77%• |
| VALEN | 90.36±0.15%• | 95.37±0.97%• | 85.48±0.62%• | 62.96±0.96%• | 37.14±0.21%• |
| PLCR | 90.01±0.59%• | 95.25±0.63%• | 86.37±0.38%• | 64.12±0.23%• | 24.59±1.68%• |
| PICO | 88.24±0.36%• | 94.89±0.46%• | 86.16±0.21%• | 62.98±0.38%• | 29.95±0.48%• |
| CAVL | 87.81±1.27%• | 93.44±1.45%• | 59.67±3.30%• | 52.59±1.01%• | 28.10±0.77%• |
| LWS | 88.79±0.34%• | 92.67±1.56%• | 37.49±2.82%• | 53.98±0.99%• | 27.37±0.82%• |
| RC | 89.52±0.65%• | 93.88±0.74%• | 85.95±0.40%• | 63.41±0.56%• | 35.74±0.61%• |
| CC | 89.78±0.48%• | 93.83±0.22%• | 79.96±0.99%• | 62.40±0.84%• | 31.46±1.24%• |
| PRODEN | 89.68±0.55%• | 93.60±0.61%• | 86.04±0.21%• | 62.56±1.49%• | 33.37±0.97%• |

way, the $j$-th branch $\varphi(\cdot; \boldsymbol{\Omega}_j)$ could be considered as an approximation of $\eta'(\boldsymbol{x})$ with $\tilde{\mathcal{Y}} = \{j\}$. Note that the multi-branch technique in our approach, as well as in [30] and earlier work [18], is merely a training technique to save space and time via sharing the feature extraction layer.

And $\boldsymbol{w}_i = [w_i^1, w_i^2, \ldots, w_i^c] \in \mathbb{R}^{1 \times c}$ is a weight vector output by a model $g$ parameterized by $\boldsymbol{\Gamma}$ given the instance $\boldsymbol{x}_i$, i.e.,

$$\boldsymbol{w}_i = g(\boldsymbol{x}_i; \boldsymbol{\Gamma}). \tag{11}$$

Since it is unknown which label in the candidate set $S_i$ of the instance $\boldsymbol{x}_i$ is the label interfering the correct label $y_{\boldsymbol{x}_i}$, we formulate the model $g$ as a meta-learner and learn-to-learn a weight vector to eliminate the disturbance of incorrect candidate labels and obtain the reduction-based pseudo-label $\boldsymbol{v}_i$ for training. We employ the reduction-based pseudo-labels $\mathbf{V} = [\boldsymbol{v}_1, \boldsymbol{v}_2, \ldots, \boldsymbol{v}_n]$ with each element $\boldsymbol{v}_i$ calculated from $\boldsymbol{w}_i$ to update the predictive model $f(\cdot; \boldsymbol{\Theta})$:

$$\mathcal{L}^{\text{inner}}(f(\mathbf{X}; \boldsymbol{\Theta}), \mathbf{V}) = -\frac{1}{n} \sum_{i=1}^{n} \ell(f(\boldsymbol{x}_i; \boldsymbol{\Theta}), \boldsymbol{v}_i), \tag{12}$$

Then, we assess the updated predictive model on the validation dataset $\mathcal{D}^{\text{val}} = \{(\boldsymbol{x}_i^{\text{val}}, \boldsymbol{y}_i^{\text{val}} | 1 \leq i \leq n^{\text{val}}\}$ to update the meta-learner $g(\cdot; \boldsymbol{\Gamma})$

$$\mathcal{L}^{\text{outer}}(f(\mathbf{X}^{\text{val}}; \boldsymbol{\Theta}), \mathbf{Y}^{\text{val}}) = -\frac{1}{n^{\text{val}}} \sum_{i=1}^{n^{\text{val}}} \ell(f(\boldsymbol{x}_i^{\text{val}}; \boldsymbol{\Theta}), \boldsymbol{y}_i^{\text{val}}). \tag{13}$$

Overall, the meta-learning objective can be formulated as a bi-level optimization problem as follows:

$$\boldsymbol{\Gamma}^{\star} = \arg\min_{\boldsymbol{\Gamma}} \mathcal{L}^{\text{outer}}(f(\mathbf{X}^{\text{val}}; \boldsymbol{\Theta}^{\star}(\boldsymbol{\Gamma})), \mathbf{Y}^{\text{val}})$$
$$\text{s.t.} \quad \boldsymbol{\Theta}^{\star} = \arg\min_{\boldsymbol{\Theta}} \mathcal{L}^{\text{inner}}(f(\mathbf{X}; \boldsymbol{\Theta}), \mathbf{V}), \tag{14}$$

To solve the optimization of Eq. (14), an online strategy inspired by [31] is employed to update $\boldsymbol{\Theta}$ and $\boldsymbol{\Gamma}$ through a single optimization loop, respectively, which guarantees the efficiency of the algorithm. Specifically, we shuffle the training set $\mathcal{D}$ into $K$ mini-batches. Each mini-batch contains $m$ examples, i.e., $\{(\boldsymbol{x}_i, S_i) | 1 \leq i \leq m\}$. In the step $k$ of training, we employ stochastic gradient descent (SGD) to optimize the meta-learning objective $\mathcal{L}^{\text{inner}}$ and $\mathcal{L}^{\text{outer}}$ with the loss functions for the classifier and auxiliary model $\mathcal{L}$ and $\mathcal{L}^{\text{aux}}$ on the $k$-th mini-batch.

First, as for the auxiliary model $\varphi$, we update the parameter of each branch $\boldsymbol{\Omega}_j^k$ to $\boldsymbol{\Omega}_j^{k+1}$ as follows:

$$\boldsymbol{\Omega}_j^{k+1} = \boldsymbol{\Omega}_j^k - \frac{\beta_1}{m} \sum_{i=1}^{m} \frac{\partial \ell(\varphi(\boldsymbol{z}_i; \boldsymbol{\Omega}_j^k), \boldsymbol{u}_i^j)}{\partial \boldsymbol{\Omega}_j^k}, \tag{15}$$

Table 2: Classification accuracy (mean±std) of comparing algorithms on the real-world datasets.

| Dataset | Lost | BirdSong | MSRCv2 | Soccer Player | Yahoo!News |
|---|---|---|---|---|---|
| RPLG | **81.07±0.74%** | **75.27±0.23%** | **51.65±0.65%** | **56.94±0.34%** | **68.01±0.19%** |
| DIRK | 79.24±0.63%● | 74.52±0.23%● | 48.59±0.28%● | 55.83±0.35%● | 67.65±0.32%● |
| POP | 78.57±0.45%● | 74.47±0.36%● | 45.86±0.28%● | 54.48±0.10%● | 66.38±0.07%● |
| IDGP | 77.02±0.80%● | 74.23±0.17%● | 50.45±0.47%● | 55.99±0.28%● | 66.62±0.19%● |
| VALEN | 76.87±0.86%● | 73.39±0.26%● | 49.97±0.43%● | 55.81±0.10%● | 66.26±0.13%● |
| CAVL | 75.89±0.42%● | 73.47±0.13%● | 44.73±0.96%● | 54.06±0.67%● | 65.44±0.23%● |
| LWS | 73.13±0.32%● | 51.45±0.26%● | 49.85±0.49%● | 50.24±0.45%● | 48.21±0.29%● |
| RC | 76.26±0.46%● | 69.33±0.32%● | 49.47±0.43%● | 56.02±0.59%● | 63.51±0.20%● |
| CC | 63.54±0.25%● | 69.90±0.58%● | 41.50±0.44%● | 49.07±0.36%● | 54.86±0.48%● |
| PRODEN | 76.47±0.25%● | 73.44±0.12%● | 45.10±0.16%● | 54.05±0.15%● | 66.14±0.10%● |

where $\beta_1$ is the step size. Then, after updating $\boldsymbol{\Omega}_j^k$ to $\boldsymbol{\Omega}_j^{k+1}$, we could obtain the reduction-based pseudo-label $\boldsymbol{v}_i$ by Eq. (8) and optimize the inner objective of the bi-level optimization Eq. (14):

$$\boldsymbol{\Theta}^{k+1} = \boldsymbol{\Theta}^k - \frac{\beta_2}{m} \sum_{i=1}^{m} \frac{\partial \ell(f(\boldsymbol{x}_i; \boldsymbol{\Theta}^k), \boldsymbol{v}_i)}{\partial \boldsymbol{\Theta}^k}, \tag{16}$$

where $\beta_2$ is the step size. Note that after updating $\boldsymbol{\Theta}^k$ to $\boldsymbol{\Theta}^{k+1}$ with the reduction-based pseudo-label $\boldsymbol{v}_i$, $\boldsymbol{\Theta}^{k+1}$ is dependent on the parameters $\boldsymbol{\Gamma}$ of the meta-learner $g$, i.e., $\boldsymbol{\Theta}^{k+1}(\boldsymbol{\Gamma}^k)$, which allows the updation of $\boldsymbol{\Gamma}^k$ through the loss function $\mathcal{L}^{\text{outer}}$ as follows:

$$\boldsymbol{\Gamma}^{k+1} = \boldsymbol{\Gamma}^k - \frac{\beta_3}{m} \sum_{i=1}^{m} \frac{\partial \ell(f(\boldsymbol{x}_i^{\text{val}}; \boldsymbol{\Theta}^{k+1}), \boldsymbol{y}_i^{\text{val}})}{\partial \boldsymbol{\Gamma}^k}, \tag{17}$$

where we also randomly sample $m$ examples from $\mathcal{D}^{\text{val}}$, and $\beta_3$ is the step size. Finally, we rollback the parameters of our classifier to $\boldsymbol{\Theta}^k$ and employ the pseudo-label $\boldsymbol{q}_i$ generated by Eq. (11)(8)(6) to optimize it with the same step size with Eq. (16):

$$\boldsymbol{\Theta}^{k+1} = \boldsymbol{\Theta}^k - \frac{\beta_2}{m} \sum_{i=1}^{m} \frac{\partial \ell(f(\boldsymbol{x}_i; \boldsymbol{\Theta}^k), \boldsymbol{q}_i)}{\partial \boldsymbol{\Theta}^k}. \tag{18}$$

Note that we need to rollback to $\boldsymbol{\Theta}_j^k$, making it return to the appropriate optimization path. The goal of Eq. (16) is to optimize $\boldsymbol{\Gamma}$ by further building the dependency between $\boldsymbol{\Theta}_j^{k+1}$ and the meta-learner parameters $\boldsymbol{\Gamma}$, while that of Eq. (18) is to pursue $\boldsymbol{\Theta}^\star$ for better prediction on unobserved instances, which is also the ultimate goal of the whole algorithm.

As the auxiliary model parameters $\{\boldsymbol{\Omega}_j\}_{j=1}^c$ and meta-learner parameters $\boldsymbol{\Gamma}$ are updated iteratively, the pseudo-label $\boldsymbol{q}_i$ is also refined to contribute to the optimization of the classifier $f(\cdot; \boldsymbol{\Theta})$ step by step. In this way, the performance of the predictive model continues to be improved in our approach RPLG. The algorithmic description of RPLG is presented in Algorithm 1 in Appendix A.3. In our framework RPLG, the influence of label $j$ is eliminated with the highest priority at the $j$-th branch $\phi(\cdot; \boldsymbol{\Omega}_j)$ of the auxiliary model, whose label subspace does not include label $j$. Hence, instances whose disturbing incorrect labels include label $j$ could obtain more correct reduction-based pseudo-labels by assigning more weight to the $j$-th branch when performing aggregation through the weight vector $\boldsymbol{\omega}$, which is learned by the meta-learner $g(\cdot; \boldsymbol{\Gamma})$.

## 4 Experiments

In this section, we validate the effectiveness of our proposed RPLG by conducting it on manually corrupted benchmark datasets and real-world datasets and comparing its results against DNN-based PLL algorithms. Also, we explore RPLG through ablation study, sensitivity analysis, convergence analysis, and time consumption. The implementation is based on Pytorch [27] with the GPU model NVIDIA RTX 3090. The source code is available at `https://github.com/palm-ml/rplg`.

Table 3: Classification accuracy (mean±std) for comparison against RPLG-NM.

| Dataset | RPLG | RPLG-NM |
|---|---|---|
| FMNIST | **91.41±0.13%** | 89.68±0.34%● |
| KMNIST | **96.85±0.11%** | 93.73±0.85%● |
| CIFAR-10 | **87.53±0.21%** | 85.43±0.52%● |
| CIFAR-100 | **65.03±0.21%** | 61.18±0.34%● |
| TinyImageNet | **40.74±0.64%** | 35.35±1.01%● |
| Lost | **81.07±0.74%** | 77.14±1.62%● |
| BirdSong | **75.27±0.23%** | 73.18±0.71%● |
| MSRCv2 | **51.65±0.65%** | 46.62±1.54%● |
| Soccer Player | **56.94±0.34%** | 55.48±0.65%● |
| Yahoo!News | **68.01±0.19%** | 66.82±0.13%● |

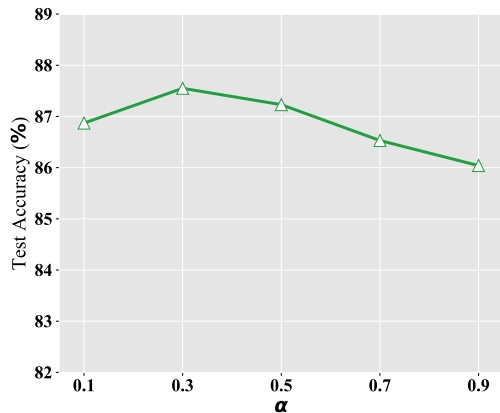

Figure 1: Sensitivity analysis of $\alpha$.

## 4.1 Datasets

RPLG and compared DNN-based PLL algorithms are implemented on three widely used benchmark datasets in deep learning: FMNIST[41], KMNIST [4], CIFAR-10, CIFAR-100 [17] and TinyImageNet [19]. For these datasets, we generate instance-dependent candidate labels through the same strategy as [44], which considers instance-dependent PLL for the first time, to create manually corrupted benchmark datasets.

Besides, since data augmentation cannot be performed on extracted features from audio and video data, our approach and data-augmentation-free PLL methods are also performed on five frequently used real-world datasets, which come from different practical application domains, including Lost [5], BirdSong [1], MSRCv2 [21], Soccer Player [50] and Yahoo!News [9].

As for benchmark datasets, we split 10% samples from the training datasets to form the validating datasets. As for real-world datasets, we conduct the algorithms with 80%/10%/10% train/validation/test split. Then we run five trials on each datasets with different random seeds and report the mean accuracy and standard deviation of all comparing algorithms.

## 4.2 Baselines

We compare RPLG with six methods, which are designed for the challenge of ID-PLL: 1) DIRK [37], a self-distillation framework which rectifies the label confidences as the distilled knowledge to guide the training of the predictive model. 2) SDCT [35], a sample selection framework which selects well-disambiguated samples based on normalized entropy in two stages for the training of the predictive model with data augmentation. 3) POP [44], a label purification framework which progressively purifies each candidate labels as the performance of the trained predictive model improves. 4) IDGP [37], a maximum-a-posterior approach which decomposes candidate labels into the results sampled from two different distributions to form a optimization objective for training. 5) ABLE [35], a contrastive learning framework which is based on data augmentation and pulls ambiguity-induced positives closer and the remaining instances further in the representation space. 6) VALEN [44], an encoder-decoder framework which leverages variational inference to recover latent label distributions for the guidance of training the model.

Besides, we also compare our method with another seven classical DNN-based PLL methods: 1) PLCR [37], a manifold regularization approach which is based on data augmentation and proposes a consistency regularization objective to preserve manifold structure in feature and label space. 2) PICO [35], a contrastive learning framework that relies on data augmentation and achieves label disambiguation through contrastive prototypes. 3) CAVL [51], a identification-based approach which identifies correct labels from candidate labels by class activation value. 4) LWS [36], an identification-based approach which introduces a leverage parameter as the trade-off between losses on candidate and non-candidate labels. 5) RC [7], a risk-consistent approach which utilizes the loss correction strategy to estimate the true risk by only using data with candidate labels. 6) CC [7], a classifier-consistent approach which leverages the transition matrix to learn a predictive model that could approximate the optimal one. 7) PRODEN [24], a self-training algorithm which normalizes the

output of the model on candidate labels and utilizes it as a weight on the cross-entropy function for training.

To ensure fairness, we utilize the same network backbone, optimizer, and data augmentation strategy across all compared methods. We take the same backbone as [42, 38] on `CIFAR-10`, `CIFAR-100` and all realworld datasets, and [20, 48] on `TinyImageNet`. The optimizer is stochastic gradient descent (SGD) [29] with momentum 0.9, batch size 256, and epoch 250.

Besides, in line with the approach presented in [37], we meticulously apply the data augmentation strategy. For hyper-parameters, we carefully select the most appropriate ones for each algorithm to ensure optimal model parameters based on their performances on the validation datasets. To mitigate overfitting, the training procedure of a model will be halted prematurely if its performance on the validation dataset fails to improve over 50 epochs.

### 4.3 Experimental Results

The performance of each DNN-based method on each corrupted benchmark dataset is summarized in Table 1, where the best results are highlighted in bold and ●/○ indicates whether RPLG statistically wins/loses to the comparing method on each dataset additionally (pairwise t-test at 0.05 significance level). Overall, we observe that RPLG significantly outperforms all comparative methods, whether ID-PLL or classic DNN-based PLL approaches, across all benchmark datasets. The improvements are especially pronounced on the complex `TinyImageNet` dataset.

Table 2 demonstrates the ability of RPLG to solve the ID-PLL problem in real-world datasets. Note that data-augmentation-based algorithms, including SDCT, ABLE, PLCR and PICO, are not compared on the real-world PLL datasets due to the inability of data augmentation to be employed on the extracted features from various domains. We can find that our method still has significantly stronger competence than others on all datasets, even the large dataset `Soccer Player` and `Yahoo!News`, against all other comparing algorithms.

### 4.4 Further Analysis

To demonstrate the effectiveness of the meta-learned weight $w$ introduced by RPLG, we explore a vanilla variant, RPLG-NM, where a uniform weight is applied to the candidate labels of the instance instead of the weight output by a parameterized model learned through meta-learning. The performance of RPLG compared to RPLG-NM is assessed using classification accuracy, with pairwise t-tests conducted at a significance level of 0.05. As shown in Table 3, by leveraging the meta-learned weight's ability to select branches without being influenced by strongly associated incorrect labels, RPLG consistently outperforms RPLG-NM across all datasets, achieving superior performance.

Also, we conduct a parameter sensitivity analysis on $\alpha$ in our algorithm, which determines the influence of pseudo-labels from the multi-branch auxiliary model. Figure 1 illustrates the sensitivity of RPLG on `CIFAR-10` as $\alpha$ increases from 0.1 to 0.9. It is evident that an $\alpha$ value around 0.3 yields superior performance for RPLG. Besides, we investigate the consistency and convergence of pseudo-labels generated by RPLG on `CIFAR-10`, as shown in Figures 2(a) and 2(b) in Appendix A.4 due to the space limit. It is clear that the generated pseudo-labels become consistent with the Bayes optimal classifier and converge as the number of epochs increases.

Furthermore, we report the training time (in hours) in Table 4, which is presented in Appendix A.4 due to the space limit. All methods were run for 250 epochs with a batch size of 256 on a single NVIDIA RTX 3090. Compared to some baselines, our approach RPLG only marginally increases the training time linearly due to meta-learning optimization and the multi-branch model. This efficiency is achieved through the online strategy employed in meta-learning optimization and shared common lower layers in the multi-branch model.

## 5 Conclusion

In this paper, we proposed a novel method of utilizing reduction-based pseudo-labels to train our predictive model by mitigating the impact of incorrect candidate labels hard to be distinguished. Reduction-based pseudo-labels are produced through weighted aggregation on the outputs of a multi-branch auxiliary model, where each branch of the model is trained within a specified label subspace.

This training strategy ensures that every branch can explicitly evade the interference from the excluded labels. Theoretically, we prove that reduction-based pseudo-labels display a higher degree of consistency with the Bayes optimal classifier.

## Acknowledgements

This research was supported by the Jiangsu Science Foundation (BG2024036, BK20243012), the National Science Foundation of China (624B2040, 62576093, 62206050, 62125602, U24A20324, and 92464301), the Fundamental Research Funds for the Central Universities (2242025K30024), and the Big Data Computing Center of Southeast University.

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

# A Technical Appendices and Supplementary Material

## A.1 Proofs of Theorem 1

*Proof.* For the conditional probability $p(\eta^\star(\boldsymbol{x}) = \arg\max_{j\in\mathcal{Y}} q^j | \boldsymbol{x} \in \mathcal{J}(\tilde{\mathcal{Y}}))$, we have:

$$
\begin{aligned}
&p(\eta^\star(\boldsymbol{x}) = \arg\max_{j\in\mathcal{Y}} q^j | \boldsymbol{x} \in \mathcal{J}(\tilde{\mathcal{Y}})) \\
&= \frac{p(\eta^\star(\boldsymbol{x}) = \arg\max_{j\in\mathcal{Y}} q^j, \boldsymbol{x} \in \mathcal{J}(\tilde{\mathcal{Y}}))}{p(\boldsymbol{x} \in \mathcal{J}(\tilde{\mathcal{Y}}))} \\
&= \frac{p(\eta^\star(\boldsymbol{x}) = \arg\max_{j\in\mathcal{Y}} q^j, \eta^{\eta^\star(\boldsymbol{x})} - \eta^a(\boldsymbol{x}) \le \tau, \boldsymbol{x} \in \mathcal{J}(\tilde{\mathcal{Y}}))}{p(\boldsymbol{x} \in \mathcal{J}(\tilde{\mathcal{Y}}))} \\
&\quad + \frac{p(\eta^\star(\boldsymbol{x}) = \arg\max_{j\in\mathcal{Y}} q^j, \eta^{\eta^\star(\boldsymbol{x})} - \eta^a(\boldsymbol{x}) > \tau, \boldsymbol{x} \in \mathcal{J}(\tilde{\mathcal{Y}}))}{p(\boldsymbol{x} \in \mathcal{J}(\tilde{\mathcal{Y}}))}
\end{aligned}
\tag{19}
$$

Since, for $\boldsymbol{x} \in \mathcal{J}(\tilde{\mathcal{Y}}), \forall j \in \tilde{\mathcal{Y}}, \eta^{\eta^\star(\boldsymbol{x})} - \eta^j(\boldsymbol{x}) \le \tau$, we could obtain:

$$
\begin{aligned}
&p(\eta^\star(\boldsymbol{x}) = \arg\max_{j\in\mathcal{Y}} q^j | \boldsymbol{x} \in \mathcal{J}(\tilde{\mathcal{Y}})) \\
&= \frac{p(\eta^\star(\boldsymbol{x}) = \arg\max_{j\in\mathcal{Y}} q^j, \eta^{\eta^\star(\boldsymbol{x})} - \eta^a(\boldsymbol{x}) \le \tau, \boldsymbol{x} \in \mathcal{J}(\tilde{\mathcal{Y}}))}{p(\boldsymbol{x} \in \mathcal{J}(\tilde{\mathcal{Y}}))} \\
&\quad + \frac{p(\eta^\star(\boldsymbol{x}) = \arg\max_{j\in\mathcal{Y}} q^j, \eta^{\eta^\star(\boldsymbol{x})} - \eta^a(\boldsymbol{x}) > \tau, \boldsymbol{x} \in \mathcal{J}(\tilde{\mathcal{Y}}))}{p(\boldsymbol{x} \in \mathcal{J}(\tilde{\mathcal{Y}}))} \\
&= \frac{p(\eta^\star(\boldsymbol{x}) = \arg\max_{j\in\mathcal{Y}} q^j, \eta^{\eta^\star(\boldsymbol{x})} - \eta^a(\boldsymbol{x}) \le \tau, \boldsymbol{x} \in \mathcal{J}(\tilde{\mathcal{Y}}))}{p(\boldsymbol{x} \in \mathcal{J}(\tilde{\mathcal{Y}}))}
\end{aligned}
\tag{20}
$$

Recall that $s = \arg\max_{j\in\mathcal{Y}, j\neq\eta^\star(\boldsymbol{x})} \eta^j(\boldsymbol{x})$. since $\forall j \in \mathcal{Y}$ with $j \neq \eta^\star(\boldsymbol{x})$. We have

$$
\eta^{\eta^\star(\boldsymbol{x})} - \eta^s(\boldsymbol{x}) \le \eta^{\eta^\star(\boldsymbol{x})} - \eta^j(\boldsymbol{x}).
\tag{21}
$$

Recall from Theorem 1 that $a = \arg\max_{j\in\tilde{\mathcal{Y}}} \eta^j(\boldsymbol{x})$, so $a \in \tilde{\mathcal{Y}}$ by definition. Additionally, since $\mathcal{J}(\tilde{\mathcal{Y}})$ consists of inputs where all $j \in \tilde{\mathcal{Y}}$ are "incorrect labels" (i.e., $y \neq j$ with $y = \eta^\star(\boldsymbol{x})$), it follows that $a \neq \eta^\star(\boldsymbol{x})$. Thus, $a \in \mathcal{Y} \setminus \{\eta^\star(\boldsymbol{x})\}$. Then, by definition, $s = \arg\max_{j\in\mathcal{Y}, j\neq\eta^\star(\boldsymbol{x})} \eta^j(\boldsymbol{x})$, meaning $s$ is the index of the maximum $\eta^j(\boldsymbol{x})$ among all labels except the true label $\eta^\star(\boldsymbol{x})$. Now, consider the structure of $\mathcal{J}(\tilde{\mathcal{Y}})$: all $j \notin \{y\} \cup \tilde{\mathcal{Y}}$ are not disturbing incorrect labels, which implies their corresponding $\eta^j(\boldsymbol{x})$ values are sufficiently small (otherwise, they would qualify as disturbing). Since $a \in \tilde{\mathcal{Y}}$ is the maximum of $\eta^j(\boldsymbol{x})$ over $\tilde{\mathcal{Y}}$, and all other labels outside $\tilde{\mathcal{Y}}$ (but in $\mathcal{Y} \setminus \{\eta^\star(\boldsymbol{x})\}$) have smaller $\eta^j(\boldsymbol{x})$, $a$ must be the maximum of $\eta^j(\boldsymbol{x})$ over the entire set $\mathcal{Y} \setminus \{\eta^\star(\boldsymbol{x})\}$. Hence, $s = a$.

Then we could obtain:

$$
\begin{aligned}
&p(\eta^\star(\boldsymbol{x}) = \arg\max_{j\in\mathcal{Y}} q^j | \boldsymbol{x} \in \mathcal{J}(\tilde{\mathcal{Y}})) \\
&= \frac{p(\eta^\star(\boldsymbol{x}) = \arg\max_{j\in\mathcal{Y}} q^j, \eta^{\eta^\star(\boldsymbol{x})} - \eta^a(\boldsymbol{x}) \le \tau, \boldsymbol{x} \in \mathcal{J}(\tilde{\mathcal{Y}}))}{p(\boldsymbol{x} \in \mathcal{J}(\tilde{\mathcal{Y}}))} \\
&= \frac{p(\eta^\star(\boldsymbol{x}) = \arg\max_{j\in\mathcal{Y}} q^j, \eta^{\eta^\star(\boldsymbol{x})} - \eta^s(\boldsymbol{x}) \le \tau, \boldsymbol{x} \in \mathcal{J}(\tilde{\mathcal{Y}}))}{p(\boldsymbol{x} \in \mathcal{J}(\tilde{\mathcal{Y}}))} \\
&= \frac{p(\eta^\star(\boldsymbol{x}) = \arg\max_{j\in\mathcal{Y}} q^j, \eta^{\eta^\star(\boldsymbol{x})} - \eta^s(\boldsymbol{x}) \le \tau, \eta^{\eta^\star(\boldsymbol{x})} - \eta^s(\boldsymbol{x}) \le 2\epsilon, \boldsymbol{x} \in \mathcal{J}(\tilde{\mathcal{Y}}))}{p(\boldsymbol{x} \in \mathcal{J}(\tilde{\mathcal{Y}}))} \\
&\quad + \frac{p(\eta^\star(\boldsymbol{x}) = \arg\max_{j\in\mathcal{Y}} q^j, \eta^{\eta^\star(\boldsymbol{x})} - \eta^s(\boldsymbol{x}) \le \tau, \eta^{\eta^\star(\boldsymbol{x})} - \eta^s(\boldsymbol{x}) > 2\epsilon, \boldsymbol{x} \in \mathcal{J}(\tilde{\mathcal{Y}}))}{p(\boldsymbol{x} \in \mathcal{J}(\tilde{\mathcal{Y}}))}
\end{aligned}
\tag{22}
$$

Since $\tau \leq 2\epsilon$, we could obtain:

$$
\begin{aligned}
&p(\eta^\star(\boldsymbol{x}) = \arg\max_{j \in \mathcal{Y}} q^j | \boldsymbol{x} \in \mathcal{J}(\tilde{\mathcal{Y}})) \\
&= \frac{p(\eta^\star(\boldsymbol{x}) = \arg\max_{j \in \mathcal{Y}} q^j, \eta^{\eta^\star(\boldsymbol{x})} - \eta^s(\boldsymbol{x}) \leq \tau, \eta^{\eta^\star(\boldsymbol{x})} - \eta^s(\boldsymbol{x}) \leq 2\epsilon, \boldsymbol{x} \in \mathcal{J}(\tilde{\mathcal{Y}}))}{p(\boldsymbol{x} \in \mathcal{J}(\tilde{\mathcal{Y}}))} \\
&\quad + \frac{p(\eta^\star(\boldsymbol{x}) = \arg\max_{j \in \mathcal{Y}} q^j, \eta^{\eta^\star(\boldsymbol{x})} - \eta^s(\boldsymbol{x}) \leq \tau, \eta^{\eta^\star(\boldsymbol{x})} - \eta^s(\boldsymbol{x}) > 2\epsilon, \boldsymbol{x} \in \mathcal{J}(\tilde{\mathcal{Y}}))}{p(\boldsymbol{x} \in \mathcal{J}(\tilde{\mathcal{Y}}))} \\
&= \frac{p(\eta^\star(\boldsymbol{x}) = \arg\max_{j \in \mathcal{Y}} q^j, \eta^{\eta^\star(\boldsymbol{x})} - \eta^s(\boldsymbol{x}) \leq \tau, \eta^{\eta^\star(\boldsymbol{x})} - \eta^s(\boldsymbol{x}) \leq 2\epsilon, \boldsymbol{x} \in \mathcal{J}(\tilde{\mathcal{Y}}))}{p(\boldsymbol{x} \in \mathcal{J}(\tilde{\mathcal{Y}}))} \\
&\leq \frac{p(\eta^\star(\boldsymbol{x}) = \arg\max_{j \in \mathcal{Y}} q^j, \eta^{\eta^\star(\boldsymbol{x})} - \eta^s(\boldsymbol{x}) \leq 2\epsilon, \boldsymbol{x} \in \mathcal{J}(\tilde{\mathcal{Y}}))}{p(\boldsymbol{x} \in \mathcal{J}(\tilde{\mathcal{Y}}))} \\
&\leq \frac{p(\eta^{\eta^\star(\boldsymbol{x})} - \eta^s(\boldsymbol{x}) \leq 2\epsilon, \boldsymbol{x} \in \mathcal{J}(\tilde{\mathcal{Y}}))}{p(\boldsymbol{x} \in \mathcal{J}(\tilde{\mathcal{Y}}))}
\end{aligned}
\tag{23}
$$

Recall that $\epsilon' \in (0, \min\{1, \min_{\boldsymbol{x} \in \mathcal{J}(\tilde{\mathcal{Y}})} \frac{(\eta^{\eta^\star(\boldsymbol{x})}(\boldsymbol{x}) - \eta^b(\boldsymbol{x}))(\eta^{\eta^\star(\boldsymbol{x})}(\boldsymbol{x}) - \eta^a(\boldsymbol{x}))}{4\epsilon(1 - \sum_{j \in \tilde{\mathcal{Y}}} \eta^j(\boldsymbol{x}))}\})$ with $a = \arg\max_{j \in \tilde{\mathcal{Y}}} \eta^j(\boldsymbol{x})$ and $b = \arg\max_{j \notin \{y\} \cup \tilde{\mathcal{Y}}} \eta^j(\boldsymbol{x})$. We have

$$
\begin{aligned}
\frac{(\eta^{\eta^\star(\boldsymbol{x})}(\boldsymbol{x}) - \eta^b(\boldsymbol{x}))(\eta^{\eta^\star(\boldsymbol{x})}(\boldsymbol{x}) - \eta^a(\boldsymbol{x}))}{2\epsilon(1 - \sum_{j \in \tilde{\mathcal{Y}}} \eta^j(\boldsymbol{x}))} &\geq 2\epsilon' \\
\frac{(\eta^{\eta^\star(\boldsymbol{x})}(\boldsymbol{x}) - \eta^b(\boldsymbol{x}))(\eta^{\eta^\star(\boldsymbol{x})}(\boldsymbol{x}) - \eta^s(\boldsymbol{x}))}{2\epsilon(1 - \sum_{j \in \tilde{\mathcal{Y}}} \eta^j(\boldsymbol{x}))} &\geq 2\epsilon' \\
\frac{(\eta^{\eta^\star(\boldsymbol{x})}(\boldsymbol{x}) - \eta^b(\boldsymbol{x}))2\epsilon}{2\epsilon(1 - \sum_{j \in \tilde{\mathcal{Y}}} \eta^j(\boldsymbol{x}))} &\geq 2\epsilon' \\
\frac{\eta^{\eta^\star(\boldsymbol{x})}(\boldsymbol{x}) - \eta^b(\boldsymbol{x})}{1 - \sum_{j \in \tilde{\mathcal{Y}}} \eta^j(\boldsymbol{x})} &\geq 2\epsilon'
\end{aligned}
\tag{24}
$$

Here, according to Eq. (1), we have $\eta'^{\eta'^\star(\boldsymbol{x})} - \eta'^s(\boldsymbol{x}) > 2\epsilon'$, and obtain:

$$
\begin{aligned}
&p(\eta^\star(\boldsymbol{x}) = \arg\max_{j \in \mathcal{Y}} q^j | \boldsymbol{x} \in \mathcal{J}(\tilde{\mathcal{Y}})) \\
&\leq \frac{p(\eta^{\eta^\star(\boldsymbol{x})} - \eta^s(\boldsymbol{x}) \leq 2\epsilon, \boldsymbol{x} \in \mathcal{J}(\tilde{\mathcal{Y}}))}{p(\boldsymbol{x} \in \mathcal{J}(\tilde{\mathcal{Y}}))} \\
&\leq \frac{p(\eta'^{\eta'^\star(\boldsymbol{x})} - \eta'^s(\boldsymbol{x}) > 2\epsilon', \boldsymbol{x} \in \mathcal{J}(\tilde{\mathcal{Y}}))}{p(\boldsymbol{x} \in \mathcal{J}(\tilde{\mathcal{Y}}))}
\end{aligned}
\tag{25}
$$

Recall that $\forall j \in \mathcal{Y} \setminus \tilde{\mathcal{Y}}, |\varphi^j(\boldsymbol{x}) - \eta'^j(\boldsymbol{x})| \leq \epsilon'$. We have

$$
\begin{aligned}
\eta'^{\eta'^\star(\boldsymbol{x})} - \eta'^s(\boldsymbol{x}) &> 2\epsilon' \\
\eta'^{\eta'^\star(\boldsymbol{x})} - \epsilon' &> \eta'^s(\boldsymbol{x}) + \epsilon' \\
\eta'^{\eta'^\star(\boldsymbol{x})} - \epsilon' &> \eta'^j(\boldsymbol{x}) + \epsilon', \forall j \in \mathcal{Y}, j \neq \eta'^\star(\boldsymbol{x}) \\
\varphi^{\eta'^\star}(\boldsymbol{x}) &> \varphi^j(\boldsymbol{x}), \forall j \in \mathcal{Y}, j \neq \eta'^\star(\boldsymbol{x})
\end{aligned}
\tag{26}
$$

Take $\arg\max_{j \in \mathcal{Y}} q^j = \arg\max_{j \in \mathcal{Y}} \varphi^j(\boldsymbol{x})$, we could obtain $\eta^\star(\boldsymbol{x}) = \arg\max_{j \in \mathcal{Y}} q'^j$.

Finally, we have

$$p(\eta^\star(\boldsymbol{x}) = \arg\max_{j\in\mathcal{Y}} q^j | \boldsymbol{x} \in \mathcal{J}(\tilde{\mathcal{Y}}))$$

$$= \frac{p(\eta^\star(\boldsymbol{x}) = \arg\max_{j\in\mathcal{Y}} q^j, \boldsymbol{x} \in \mathcal{J}(\tilde{\mathcal{Y}}))}{p(\boldsymbol{x} \in \mathcal{J}(\tilde{\mathcal{Y}}))}$$

$$= \frac{p(\eta^\star(\boldsymbol{x}) = \arg\max_{j\in\mathcal{Y}} q^j, \eta^{\eta^\star(\boldsymbol{x})} - \eta^a(\boldsymbol{x}) \le \tau, \boldsymbol{x} \in \mathcal{J}(\tilde{\mathcal{Y}}))}{p(\boldsymbol{x} \in \mathcal{J}(\tilde{\mathcal{Y}}))}$$

$$\quad + \frac{p(\eta^\star(\boldsymbol{x}) = \arg\max_{j\in\mathcal{Y}} q^j, \eta^{\eta^\star(\boldsymbol{x})} - \eta^a(\boldsymbol{x}) > \tau, \boldsymbol{x} \in \mathcal{J}(\tilde{\mathcal{Y}}))}{p(\boldsymbol{x} \in \mathcal{J}(\tilde{\mathcal{Y}}))}$$

$$= \frac{p(\eta^\star(\boldsymbol{x}) = \arg\max_{j\in\mathcal{Y}} q^j, \eta^{\eta^\star(\boldsymbol{x})} - \eta^a(\boldsymbol{x}) \le \tau, \boldsymbol{x} \in \mathcal{J}(\tilde{\mathcal{Y}}))}{p(\boldsymbol{x} \in \mathcal{J}(\tilde{\mathcal{Y}}))}$$

$$= \frac{p(\eta^\star(\boldsymbol{x}) = \arg\max_{j\in\mathcal{Y}} q^j, \eta^{\eta^\star(\boldsymbol{x})} - \eta^s(\boldsymbol{x}) \le \tau, \boldsymbol{x} \in \mathcal{J}(\tilde{\mathcal{Y}}))}{p(\boldsymbol{x} \in \mathcal{J}(\tilde{\mathcal{Y}}))}$$

$$= \frac{p(\eta^\star(\boldsymbol{x}) = \arg\max_{j\in\mathcal{Y}} q^j, \eta^{\eta^\star(\boldsymbol{x})} - \eta^s(\boldsymbol{x}) \le \tau, \eta^{\eta^\star(\boldsymbol{x})} - \eta^s(\boldsymbol{x}) \le 2\epsilon, \boldsymbol{x} \in \mathcal{J}(\tilde{\mathcal{Y}}))}{p(\boldsymbol{x} \in \mathcal{J}(\tilde{\mathcal{Y}}))} \tag{27}$$

$$\quad + \frac{p(\eta^\star(\boldsymbol{x}) = \arg\max_{j\in\mathcal{Y}} q^j, \eta^{\eta^\star(\boldsymbol{x})} - \eta^s(\boldsymbol{x}) \le \tau, \eta^{\eta^\star(\boldsymbol{x})} - \eta^s(\boldsymbol{x}) > 2\epsilon, \boldsymbol{x} \in \mathcal{J}(\tilde{\mathcal{Y}}))}{p(\boldsymbol{x} \in \mathcal{J}(\tilde{\mathcal{Y}}))}$$

$$= \frac{p(\eta^\star(\boldsymbol{x}) = \arg\max_{j\in\mathcal{Y}} q^j, \eta^{\eta^\star(\boldsymbol{x})} - \eta^s(\boldsymbol{x}) \le \tau, \eta^{\eta^\star(\boldsymbol{x})} - \eta^s(\boldsymbol{x}) \le 2\epsilon, \boldsymbol{x} \in \mathcal{J}(\tilde{\mathcal{Y}}))}{p(\boldsymbol{x} \in \mathcal{J}(\tilde{\mathcal{Y}}))}$$

$$\le \frac{p(\eta^\star(\boldsymbol{x}) = \arg\max_{j\in\mathcal{Y}} q^j, \eta^{\eta^\star(\boldsymbol{x})} - \eta^s(\boldsymbol{x}) \le 2\epsilon, \boldsymbol{x} \in \mathcal{J}(\tilde{\mathcal{Y}}))}{p(\boldsymbol{x} \in \mathcal{J}(\tilde{\mathcal{Y}}))}$$

$$\le \frac{p(\eta^{\eta^\star(\boldsymbol{x})} - \eta^s(\boldsymbol{x}) \le 2\epsilon, \boldsymbol{x} \in \mathcal{J}(\tilde{\mathcal{Y}}))}{p(\boldsymbol{x} \in \mathcal{J}(\tilde{\mathcal{Y}}))}$$

$$\le \frac{p(\eta'^{\eta'^\star(\boldsymbol{x})} - \eta'^s(\boldsymbol{x}) > 2\epsilon', \boldsymbol{x} \in \mathcal{J}(\tilde{\mathcal{Y}}))}{p(\boldsymbol{x} \in \mathcal{J}(\tilde{\mathcal{Y}}))}$$

$$\le \frac{p(\eta^\star(\boldsymbol{x}) = \arg\max_{j\in\mathcal{Y}} q'^j, \boldsymbol{x} \in \mathcal{J}(\tilde{\mathcal{Y}}))}{p(\boldsymbol{x} \in \mathcal{J}(\tilde{\mathcal{Y}}))}$$

$$= p(\eta^\star(\boldsymbol{x}) = \arg\max_{j\in\mathcal{Y}} q'^j | \boldsymbol{x} \in \mathcal{J}(\tilde{\mathcal{Y}}))$$

The proof has been completed.

## A.2 Proofs of Theorem 2

$$p(\eta^\star(\boldsymbol{x}) = \arg\max_{j\in\mathcal{Y}} q'^j | \boldsymbol{x} \in \mathcal{J}(\tilde{\mathcal{Y}}))$$

$$\ge p(\eta'^{\eta'^\star(\boldsymbol{x})} - \eta'^s(\boldsymbol{x}) > 2\epsilon' | \boldsymbol{x} \in \mathcal{J}(\tilde{\mathcal{Y}}))$$

$$\ge p(\eta^{\eta^\star(\boldsymbol{x})} - \eta^s(\boldsymbol{x}) > \frac{4\epsilon\epsilon'(1-\eta^s(\boldsymbol{x}))}{\eta^{\eta^\star(\boldsymbol{x})} - \eta^b(\boldsymbol{x})} | \boldsymbol{x} \in \mathcal{J}(\tilde{\mathcal{Y}})) \tag{28}$$

$$= 1 - p(\eta^{\eta^\star(\boldsymbol{x})} - \eta^s(\boldsymbol{x}) \le \frac{4\epsilon\epsilon'(1-\eta^s(\boldsymbol{x}))}{\eta^{\eta^\star(\boldsymbol{x})} - \eta^b(\boldsymbol{x})} | \boldsymbol{x} \in \mathcal{J}(\tilde{\mathcal{Y}}))$$

Since for $\boldsymbol{x} \in \mathcal{J}(\tilde{\mathcal{Y}})$, its posterior $\eta(\boldsymbol{x})$ fulfills Assumption 1 for constants $C, \lambda > 0$ and $t_0 \in (0, 1]$, we have

$$p(\eta^\star(\boldsymbol{x}) = \arg\max_{j\in\mathcal{Y}} q'^j | \boldsymbol{x} \in \mathcal{J}(\tilde{\mathcal{Y}})) \ge 1 - C(\frac{4\epsilon\epsilon'(1-\eta^s(\boldsymbol{x}))}{\eta^{\eta^\star(\boldsymbol{x})} - \eta^b(\boldsymbol{x})})^\lambda = 1 - C[O(\epsilon\epsilon')]^\lambda \tag{29}$$

The proof has been completed.

---

**Algorithm 1** RPLG Algorithm

---

**Input:** PLL training dataset $\mathcal{D} = \{(\boldsymbol{x}_i, S_i | 1 \leq i \leq n\}$, validating dataset $\mathcal{D}^{\text{val}} = \{(\boldsymbol{x}_i^{\text{val}}, \boldsymbol{y}_i^{\text{val}} | 1 \leq i \leq n^{\text{val}}\}$, Epoch $I$, Iteration $K$;

1: Initialize the parameters of the predictive model, the auxiliary model and meta-learner, i.e., $\boldsymbol{\Theta}^0$, $\{\boldsymbol{\Omega}_j^0\}_{j=1}^c$ and $\boldsymbol{\Gamma}^0$;
2: **for** $i = 1, 2, \ldots, I$ **do**
3:     Randomly shuffle the training dataset $\mathcal{D}$ and divide it into $K$ mini-batches;
4:     **for** $k = 0, 1, \ldots, K-1$ **do**
5:         Calculate $\mathbf{U}_i$ for the instance $\boldsymbol{x}_i$ according to Eq. (9);
6:         Update $\boldsymbol{\Omega}_j^k$ to $\boldsymbol{\Omega}_j^{k+1}$ according to Eq. (15);
7:         Calculate $\boldsymbol{v}_i$ for the instance $\boldsymbol{x}_i$ according to Eq. (8);
8:         Save $\boldsymbol{\Theta}_j^k$ and update $\boldsymbol{\Theta}_j^k$ to $\boldsymbol{\Theta}_j^{k+1}$ according to Eq. (16);
9:         Randomly sample a mini-batch from $\mathcal{D}^{\text{val}}$ and update $\boldsymbol{\Gamma}_j^k$ to $\boldsymbol{\Gamma}_j^{k+1}$ according to Eq. (17);
10:        Calculate $\boldsymbol{q}_i$ for the instance $\boldsymbol{x}_i$ according to Eq. Eq. (11)(8)(6);
11:        Rollback to $\boldsymbol{\Theta}_j^k$ and update $\boldsymbol{\Theta}_j^k$ to $\boldsymbol{\Theta}_j^{k+1}$ according to Eq. (18);
12:     **end for**
13: **end for**
**Output:** The predictive model $f(\cdot; \boldsymbol{\Theta})$.

---

### A.3 Algorithm Table

The algorithmic description of RPLG is presented in Algorithm 1. The reduction-based pseudo-labels are aggregated from the outputs of an auxiliary multi-branch model. Specifically, each branch of this auxiliary model is trained within a distinct label subspace that explicitly excludes a set of specific labels. To generate the final reduction-based pseudo-label, we leverage a meta-learned weight vector to fuse the output results of all these branches. Trained with the reduction-based pseudo-labels, the predictive performance of the model is consistently enhanced in our proposed approach RPLG.

### A.4 Further Analysis

**Consistency and Convergence.** We investigate the consistency and convergence of pseudo-labels generated by our method RPLG on the `CIFAR-10` dataset, with results presented in Figures 2(a) and 2(b). As evident from these figures, the generated pseudo-labels gradually align with the Bayes optimal classifier and exhibit convergence as the number of training epochs increases.

**Time Consumption Analysis.** Table 4 presents training time measured in hours. All the methods were executed for 250 epochs with a batch size of 256 on a single NVIDIA RTX 3090 with AMD EPYC 7453 28-Core Processor. In contrast to recent baseline methods, our proposed approach, RPLG experiences only linear increase in training time. The efficiency is attained by virtue of the online strategy used in meta-learning optimization and shared lower layers in the multi-branch model.

**Trade-off between Effectiveness and Efficiency.** One-branch-per-label is a universal and conservative scheme to deal with ID-PLL, since each class may act as a disturbing class for other classes. In practice, when efficiency is preferred, using fewer branches via label clustering or dimensionality reduction is a good idea. Here, we formulate two variants of RPLG with fewer branches: RPLG-W and RPLG-R. RPLG-W employs Word2Vec for label clustering to group labels into a cluster with similar textual semantics (e.g., "dog" and "cat"). It retains only those branches where the excluded label belongs to the clusterthis is because labels with similar semantics are more likely to interfere with each other, so training a series of subspaces that mutually exclude semantically similar labels is helpful. In contrast, RPLG-R retains branches randomly. Table 5 presents the performance of these two variants on TinyImageNet. From Table 5, we can observe that compared with randomly retaining branches, preserving branches that exclude semantically similar labels achieves, to some extent, a favorable trade-off between performance and computational overhead.

Table 4: Comparison of the training time consumed by all approaches on `CIFAR10`, `CIFAR-100` and `TinyImageNet`.

| Method | CIFAR10 | CIFAR100 | TinyImageNet |
|--------|---------|----------|--------------|
| RPLG | 2.32 | 3.01 | 6.38 |
| DIRK | 1.50 | 2.11 | 3.23 |
| SDCT | 1.86 | 1.93 | 6.15 |
| POP | 1.48 | 1.51 | 5.84 |
| IDGP | 2.91 | 2.90 | 6.34 |
| ABLE | 3.03 | 3.06 | 6.49 |
| VALEN | 1.18 | 1.37 | 5.63 |
| PLCR | 1.38 | 1.44 | 5.27 |
| PICO | 3.12 | 3.15 | 5.96 |
| CAVL | 1.51 | 1.53 | 5.53 |
| LWS | 1.66 | 1.67 | 5.88 |
| RC | 1.47 | 1.50 | 5.55 |
| CC | 1.41 | 1.43 | 5.16 |
| PRODEN | 1.42 | 1.43 | 5.14 |

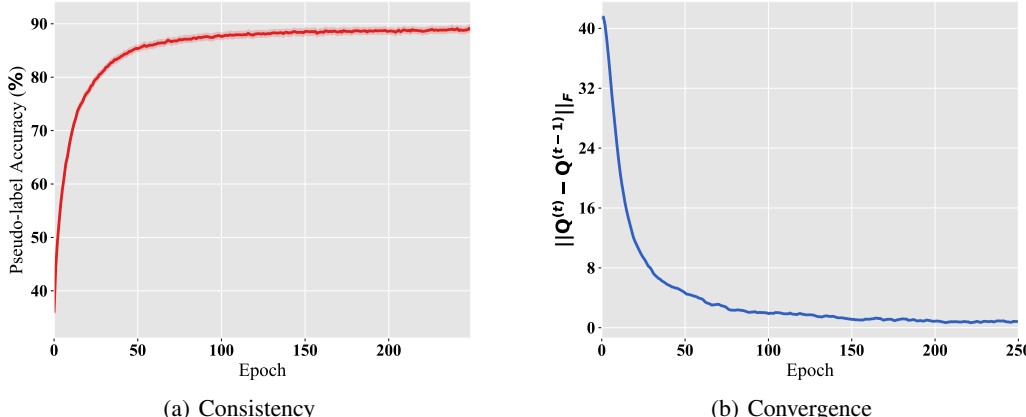

(a) Consistency

(b) Convergence

Figure 2: Further analysis of RPLG on `CIFAR-10`

## A.5 Limitations and Future Work

This is the first work to introduce the concept of reduction-based pseudo-labels. This pseudo-label generation approach is motivated by derived theorems and is appropriate for addressing the challenge of instance-dependent partial label learning and the limitations of previous works in this area. However, training the pseudo-label generation model in the label subspace does not necessarily require all training samples, as some samples outside the subspace may act as noise for model training within the subspace. In future work, we will introduce a sample selection mechanism for training the subspace pseudo-label generation model, which is expected to improve the model's training accuracy and efficiency.

## A.6 Impact Statement

The aim of this study is to advance the techniques and methodologies in the field of Machine Learning. The approach we propose to deal with ID-PLL, a typically weakly supervised learning, may bring about a situation where data annotators or other personnel involved in data-related occupations could potentially be replaced. We are acutely conscious of the importance of addressing the impacts of automation on employment and are vigilant about its societal ramifications.

Table 5: Classification accuracy and time consumption of our approach RPLG and its variants RPLG-W and RPLG-R with various branch retaining rates on TinyImageNet.

| Method | Retaining Rate | Accuracy(%) | Time(h) |
|---|---|---|---|
| RPLG | 100% | 40.74 | 6.38 |
| RPLG-W | 80% | 40.63 | 6.30 |
| | 60% | 40.27 | 6.23 |
| | 40% | 39.89 | 6.09 |
| | 20% | 39.53 | 5.97 |
| RPLG-R | 80% | 39.65 | 6.30 |
| | 60% | 39.05 | 6.23 |
| | 40% | 38.78 | 6.09 |
| | 20% | 38.84 | 5.97 |
| POP (ranked second) | - | 39.27 | 5.84 |

