# OpenReview forum: "Reduction-based Pseudo-label Generation for Instance-dependent Partial Label Learning"
_NeurIPS.cc/2025/Conference — NeurIPS 2025 poster_

### Official Review · Reviewer_dbq6 · 2025-06-03

**Clarity:** 2
**Significance:** 3
**Originality:** 3
**Rating:** 4
**Confidence:** 4

**Summary:**

This paper addresses instance-dependent partial label learning (ID-PLL) by proposing RPLG, a method that generates reduction-based pseudo-labels using a multi-branch auxiliary model. The key insight is that some incorrect candidate labels strongly correlated with features are difficult to distinguish from correct labels, creating a bottleneck in training. To address this, each branch of the auxiliary model is trained in a label subspace excluding certain labels, allowing instances affected by excluded labels to benefit from unaffected branches. The authors provide theoretical analysis showing that reduction-based pseudo-labels can achieve better consistency with the Bayes optimal classifier, and demonstrate consistent improvements across multiple benchmark and real-world datasets.

**Questions:**

1. Beyond α, how sensitive is the method to other hyperparameters in the bi-level optimization? The paper mentions "carefully selected" parameters but provides limited guidance.
2. How do you handle cases where auxiliary branches produce worse pseudo-labels than the original candidate sets? Is there any mechanism to detect and mitigate such degradation? What's the failure rate of pseudo-label generation across different datasets?
3. Since branches share feature extraction layers to save computation, how do you ensure that the learned representations don't carry interference from excluded labels? Wouldn't this shared representation potentially undermine the core premise of label subspace training?

**Ethical Concerns:**

["NO or VERY MINOR ethics concerns only"]

**Final Justification:**

I have reviewed the authors' rebuttal and discussions with other reviewers. The main concerns have been adequately addressed, and the authors provided convincing responses to the technical questions raised. I maintain my borderline accept recommendation.

**Limitations:**

The requirement for training c auxiliary branches makes the approach prohibitively expensive for datasets with hundreds or thousands of classes, limiting its applicability to large-scale modern applications. Besides, I don't believe that the paper has a potential negative social impact.

**Paper Formatting Concerns:**

No major formatting concerns identified.

**Quality:**

3

**Strengths And Weaknesses:**

Strengths：
1. The identification of "disturbing incorrect labels" that are strongly correlated with features represents a genuine challenge in ID-PLL that existing methods overlook.
2. The reduction-based pseudo-label generation using multi-branch auxiliary models is creative and theoretically motivated, offering a principled way to handle label interference.
3. Theorems 1 and 2 provide theoretical justification showing that reduction-based pseudo-labels can achieve better consistency with the Bayes optimal classifier under certain conditions.

Weaknesses：
1. The method directly uses generated pseudo-labels without any confidence estimation or quality control mechanism. Poor-quality pseudo-labels from unreliable branches could introduce more noise than the original candidate labels, but there's no safeguard against this degradation.
2. Defining subspaces by simply excluding individual labels ignores semantic relationships between labels. Labels with similar meanings or hierarchical relationships should be considered together when defining exclusion sets. The current approach treats all labels as independent, which may not reflect real-world label structures.
3. The relationship between the "disturbing incorrect labels" concept and the practical multi-branch implementation could be clearer

---

> ### Author Rebuttal · Authors · 2025-07-31
>
> We sincerely thank you for your careful review and constructive feedback. We are delighted by your recognition of **our identification of "disturbing incorrect labels," the creativity of reduction-based pseudo-label generation, and the robust theoretical justification**. Below, we address your Weaknesses (W), Questions (Q), and Limitations (L) with detailed Explanations (E).
>
> ---
>
> **W1/Q2**: Quality control on pseudo-labels
>
> **E1**:  While our method does not explicitly use traditional confidence scores (e.g., entropy-based thresholds), it incorporates adaptive weighting via meta-learning and hybrid pseudo-label fusion to control quality.
>
> (1) Adaptive weighting via meta-learning:
>
> > The core of our reduction-based pseudo-label generation (RPLG) lies in the **meta-learned weight vector $w_i$** in Eq. (11), which serves as a dynamic quality-control mechanism. Unlike uniform aggregation (as in our ablation RPLG-NM), $w_i$ is learned via a meta-learner $g$ to prioritize outputs from branches that are **least affected by the instance’s specific disturbing labels**.
>
> (2) Hybrid pseudo-label fusion:
>
> > The final pseudo-label $q_i$ in Eq. (6) is a weighted combination of two components:
> > - $v_i$: the reduction-based pseudo-label from the multi-branch auxiliary model, filtered via meta-learned weights.
> > - $\mu_i$: a basic pseudo-label derived from the predictive model’s outputs in the full label space, serving as **an implicit quality-control buffer**.
>
> These mechanisms collectively prevent low-quality pseudo-labels from degrading performance, as validated by consistent outperformance over baselines and ablations. As suggested, we additionally report the final failure rates of pseudo-label generation across datasets, compared with PRODEN [1], in Table 1. [1] Lv et al. Progressive Identification of True Labels for Partial-Label Learning. ICML.
>
> **Table 1.** _Failure Rate of our approach RPLG and PRODEN on benchmark datasets._
>
> |Method|FMNIST|KMNIST|CIFAR-10|CIFAR-100|TinyImageNet|
> |-|-|-|-|-|-|
> |RPLG|11.21|5.34|13.56|36.87|62.45|
> |PRODEN|14.55|6.87|14.66|39.25|67.88|
>
>
>
> ---
>
> **W2**: Ignoring semantic label relations
>
> **E2**: Please allow us to further clarify that "Individual Label Exclusion" serves as a flexible and general foundation, rather than a rigid constraint.
>
> On one hand, through adaptive weight aggregation, the current design of excluding individual labels implicitly handles semantic/hierarchical relationships. For labels with strong semantic correlations (e.g., "cat" and "kitten" in an image dataset), their disturbing effects on a given instance often overlap: if an instance is troubled by "cat," it is likely also affected by "kitten." In such cases, the meta-learner $ g $ (which outputs weight vector $ w_i $) will automatically assign higher weights to branches excluding both labels.
>
> On the other hand, when efficiency is preferred, we could retain less branches by considering semantic relationships between labels. Here, we formulate a variant RPLG-W of our approach. RPLG-W employs the Word2Vec technique for label clustering to group labels into a cluster with similar textual semantics (e.g., "dog" and "cat"). Then, it retains only those branches where the excluded label belongs to the cluster—this is because labels with similar semantics are more likely to interfere with each other, so training a series of subspaces that mutually exclude semantically similar labels is helpful.
>
> Table 2 presents the performance of RPLG and RPLG-W with fewer branches. From Table 2, we can observe that preserving branches that exclude semantically similar labels achieves a favorable trade-off between performance and computational overhead to some extent.
>
> **Table 2.** _Classification accuracy and time consuming of our approach RPLG and variants with fewer branches on TinyImageNet._
>
> |Method|Branches Retaining Rate|Average Accuracy(%)|Time(h)|
> |---|---|---|---|
> |RPLG|100%|40.74|6.38|
> |RPLG-W|80%|40.63|6.30|
> ||60%|40.27|6.23|
> ||40%|39.89|6.09|
> ||20%|39.53|5.97|
> |POP (ranked second)| - |39.27|5.84|
>
> ---
>
> **W3**: The relationship between disturbing incorrect labels and multi-branch implementation
>
> **E3**: Thank you for helping us further improve the clarity of the paper. We will add the following paragraph after Eq. (10) to strengthen the relationship between disturbing incorrect labels and multi-branch implementation.
>
> > The multi-branch implementation is a practical instantiation of the strategy to counteract disturbing incorrect labels:
> > - **Disturbing labels** define the set of problematic labels that must be excluded to reduce ambiguity;
> > - **Each branch** is a specialized module trained to avoid one such label, eliminating its interference for instances troubled by it;
> > - **Meta-learned aggregation** ensures instances rely on the branches most effective at mitigating their specific disturbing labels.
>
> This tight coupling between theory (disturbing labels) and implementation (multi-branch design) is what enables RPLG to outperform methods that ignore this targeted exclusion logic.
>
> ---
>
> **Q1**: Sensitivity of other hyperparameters in the bi-level optimization
>
> **E4**: Beyond $\alpha$, the bi-level optimization in RPLG involves Step sizes ($\beta_1, \beta_2, \beta_3$), which control the update rates for the auxiliary model. In practice, ($\beta_1, \beta_2, \beta_3$) is set to be equal to the learning rate. Hence, they are selected within the range of [1e-4, 1e-1], consistent with the learning rate. We will further clarify this point in the last paragraph of Section 4.2. Here, we also provide their individual performance on CIFAR-10 for reference in Table 3.
>
> **Table 3.** _Classification accuracy of RPLG when $\beta_1, \beta_2, \beta_3$ varies on CIFAR-10._
>
> |Hyper-parameters|1e-4|1e-3|1e-2|1e-1|
> |-|-|-|-|-|
> |$\beta_1$|82.54|83.68|**87.53**|73.24|
> |$\beta_2$|81.63|84.29|**87.53**|75.35|
> |$\beta_3$|79.46|83.97|**87.53**|77.88|
>
>
> ---
>
> **Q3**: Shared feature extractor undermining label subspace training
>
> **E5**:  Shared feature extraction layers do not carry label-specific interference because: (1) they encode generic patterns; (2) branch-specific heads and losses enforce subspace isolation. This design balances computational efficiency with the integrity of label subspace training. The shared feature layer operates similarly to pre-trained backbones in transfer learning: generic features provide a strong foundation, while task-specific heads adapt them to new objectives. In our case, "tasks" are the label subspaces, and the branch heads adapt shared features to avoid excluded labels.
>
> Additionally, we provide a variant RPLG-MLP, where the linear layer branches are replaced with MLP layers for further verification. Table 4 presents the performance of this variant and RPLG on FMNIST and CIFAR-10. From this, we could infer that the design of shared representation does not undermine the core premise of label subspace training.
>
> **Table 4.** _Classification accuracy (mean±std) of RPLG and variants with MLP branches on FMNIST and CIFAR-10._
>
> |Method|FMNIST|CIFAR-10|
> |-|-|-|
> |RPLG|91.41±0.13|87.53±0.21|
> |RPLG-MLP|91.32±0.11|87.46±0.23|
>
> ---
>
> **L1**: Large-scale application
>
> **E6**: We appreciate the reviewer’s concern about the scalability of the multi-branch design for large-scale datasets with hundreds or thousands of classes. We have already clarified this limitation and the directions for future work to improve upon it in Appendix A.5. In **E2**, we have provided a more efficient variant along with preliminary experimental results, which will be supplemented in the revised version. We believe this will further enhance its applicability in large-scale scenarios.

---

### Official Review · Reviewer_S4hy · 2025-06-26

**Clarity:** 4
**Significance:** 3
**Originality:** 4
**Rating:** 5
**Confidence:** 5

**Summary:**

This paper addresses instance-dependent partial label learning (ID-PLL), where training instances are annotated with candidate labels that may include feature-correlated incorrect labels. Existing approaches leveraging the training model's identification capability often encounter a bottleneck: within the originally given label space, the model struggles to distinguish strongly feature-correlated incorrect labels from true labels, leading to poor-quality supervision signals. To tackle this, this paper proposes reduction-based pseudo-labels generated via a multi-branch auxiliary model, each branch trained in a label subspace excluding specific labels. This design explicitly mitigates interference from hard-to-identify incorrect labels. Theoretically, it proves that reduction-based pseudo-labels exhibit higher consistency with the Bayes optimal classifier than pseudo-labels directly derived from the predictive model, validated through rigorous analysis on benchmark and real-world datasets.

**Questions:**

Please refer to the weaknesses.

**Ethical Concerns:**

["NO or VERY MINOR ethics concerns only"]

**Final Justification:**

I have read all the reviews and the authors' response. All my questions have been addressed. I keep the ratings unchanged.

**Limitations:**

Yes. The authors have adequately addressed the limitations and potential negative societal impact of their work in Appendix A.5 and A.6.

**Quality:**

4

**Strengths And Weaknesses:**

Strengths:

1. **Strong Novelty**
   - The focus on the ID-PLL learning paradigm is novel. The paper emphasizes candidate labels that are difficult to identify in the original label space, which are clearly defined as ($\tau$, $f$, $\epsilon$)-disturbing incorrect labels in the paper.
   - The paper presents novel theories by being the first to explore the posterior probability related to the Bayes classifier in label subspaces for partial label learning.
   - The algorithm designed based on this theory is novel. The core idea of leveraging a multi-branch auxiliary model—where each branch is trained in a label subspace excluding specific labels—provides an insightful solution to mitigate interference from feature-correlated incorrect labels. This design explicitly addresses the limitation of prior methods.

2. **Theoretical Rigor**
   The paper makes significant theoretical contributions by proving that reduction-based pseudo-labels generated in label subspaces exhibit stronger consistency with the Bayes optimal classifier than those derived directly from the predictive model. The theorems (Theorem 1 and 2) and their proofs (Appendix A.1–A.2) are meticulously structured, establishing a rigorous theoretical foundation for the method. This theoretical insight validates the algorithm’s superiority in handling label ambiguity.

3. **Comprehensive Empirical Validation**
   The experimental design is robust, spanning diverse benchmark datasets (e.g., FMNIST, CIFAR-100, TinyImageNet) and real-world applications (e.g., Lost, Yahoo!News). RPLG demonstrates significant performance advantages, particularly on complex datasets like TinyImageNet, highlighting its generalizability. The time complexity analysis shows that RPLG achieves state-of-the-art results with only a marginal linear increase in training time, balancing efficiency and accuracy.

4. **Clarity in Presentation**
   The paper is meticulously written and well-structured, with clear notations and detailed algorithm descriptions (e.g., Algorithm 1 in Appendix A.3). Tables and figures (e.g., Tables 1–4, Figure 2) effectively visualize results, while the ablation studies and sensitivity analysis (e.g., on $\alpha$ and meta-learned weights) enhance reproducibility.

Weaknesses:

1. It is better to further explain the objective optimization function of meta-learning in Eq. (14).

2. It is better to further highlight the advantages of the algorithm in the interpretation of the experimental results section.

---

> ### Author Rebuttal · Authors · 2025-07-31
>
> We would like to express our heartfelt gratitude for your high-level evaluation of our paper. Your recognition of **the strong novelty in the ID-PLL learning paradigm, theoretical rigor, and comprehensive empirical validation** is a great encouragement. In response to Weaknesses (W), we would like to provide the following Explanations (E).
>
> ---
>
> **W1**: Clarify meta-learning optimization
>
> **E1**: We provide the following explanation for Eq. (14):
>
> > Eq. (14) implements a bi-level optimization framework where the meta-learner optimizes weights to minimize the predictive model’s validation loss after inner-loop updates. The outer optimization updates the meta-learner parameter $\Gamma$ (which generates weight vector $w_i$) by minimizing $\mathcal{L}^{\text{outer}}$, evaluating the inner-optimized predictive model on the validation set $D^{\text{val}}$. The inner optimization updates the predictive model parameter $\Theta$ using $\mathcal{L}^{\text{inner}}$, leveraging reduction-based pseudo-labels $V$ (derived from the multi-branch auxiliary model) for training. The inner-optimized $\Theta^{\star}$ depends on the outer $\Gamma$ (as $V$ relies on $w_i$), while the outer optimization’s objective depends on the inner $\Theta^{\star}$.
>
> ---
>
> **W2**: Better highlight experimental advantages
>
> **E2**: We will revise the experimental section to emphasize key findings with enhanced statistical evidence:
>
> (1) **Superior performance on complex datasets** (TinyImageNet, Soccer Player, Yahoo!News):
> > Our algorithm outperforms the second-best baseline algorithms on large, complex datasets, achieving a 1.47% improvement on TinyImageNet compared to POP and a 0.95% improvement on Soccer Player compared to IDGP.
>
> (2) **Impact of meta-learning**:
> > By leveraging meta-learned weights to select branches while mitigating the influence of strongly associated incorrect labels, RPLG consistently outperforms RPLG-NM across all datasets, with a notable 5.39% improvement on TinyImageNet.
>
> These results will be highlighted more prominently in Section 4.3.

---

> > ### Comment · Reviewer_S4hy · 2025-08-06
> >
> > Thank you for your response. All my concerns have been addressed and I keep the scores.

---

### Official Review · Reviewer_LSa6 · 2025-06-29

**Clarity:** 4
**Significance:** 4
**Originality:** 3
**Rating:** 4
**Confidence:** 5

**Summary:**

This paper proposed to leverage reduction-based pseudo-labels to alleviate the influence of incorrect candidate labels and train the predictive model to overcome current training for Instance-dependent Partial Label Learning. The proposed method directly tackles an important gap in existing PLL research, offering a compelling motivation. In addition, the theoretical analysis is rigorous and well-integrated with the proposed methodology.

**Questions:**

Please see the weaknesses which need for more evidences and explanations.

Additionally, please answer the following questions:
1. While the method is applicable to general PLL settings, the authors should evaluate and report its performance under uniformly random candidate label selection (i.e., uniform PLL) to demonstrate broader applicability.

**Ethical Concerns:**

["NO or VERY MINOR ethics concerns only"]

**Final Justification:**

The positive (motivation, theoritical analysis, and experiments) outweights the negative (novelty). I keep positive to the paper.

**Limitations:**

The proposed method uses a meta-learner to optimize the sample-wise weight w, but only compares it against a uniform weighting strategy (RPLG-NM). Incorporating comparisons with alternative weight learning or fusion strategies (e.g., self-training, attention mechanisms, or ensemble-based reweighting) would further substantiate the claimed advantages.

**Quality:**

3

**Strengths And Weaknesses:**

**Strengths:**
1. This paper demonstrates a high standard of scholarly presentation, with clear organization and a coherent logical structure that aligns well with the expectations of top-tier conferences in the field.
2. This paper addresses a critical challenge in Instance-Dependent Partial Label Learning (ID-PLL) — distinguishing the ground-truth label from misleading candidate labels — through a novel multi-branch auxiliary model architecture. This design directly tackles an important gap in existing PLL research, offering a compelling motivation.
3. The theoretical analysis is rigorous and well-integrated with the proposed methodology. The mathematical derivations are sound and effectively support the algorithmic design.
4. The experimental evaluation is thorough and well-executed, showcasing strong empirical performance across multiple benchmark datasets. The results clearly demonstrate the effectiveness of the proposed approach.

**Weaknesses:**
1. The paper would benefit from a more in-depth discussion of the theoretical component’s significance, particularly how the theory provides insights into or strengthens the proposed method.
2. Since the multi-branch architecture employed is not entirely novel, this part of the contribution may be perceived as incremental. The authors should clarify what specific innovations differentiate their approach from prior multi-branch frameworks.
3. The hyperparameter $\alpha$ plays an important role in the proposed method. A more detailed ablation study focusing on its impact would provide deeper insight into the model's robustness and sensitivity.
4. The proposed method uses a meta-learner to optimize the sample-wise weight $w$, but only compares it against a uniform weighting strategy (RPLG-NM). Incorporating comparisons with alternative weight learning or fusion strategies (e.g., self-training, attention mechanisms, or ensemble-based reweighting) would further substantiate the claimed advantages.

---

> ### Author Rebuttal · Authors · 2025-07-31
>
> Thank you very much for your thorough review of our paper. We are grateful for your affirmation of **the high-standard scholarly presentation, the novel architecture for addressing the key challenge in ID-PLL, the rigorous theoretical analysis, and the well-executed experimental evaluation**. In response to Weaknesses (W), Questions (Q) and Limitations (L), we would like to provide the following Explanations (E).
>
> ---
>
> **W1**: Need for a deeper discussion of the theory’s significance and insight into the proposed method
>
> **E1**: Thank you for pointing this out. To underscore the theory's significance and insight into the proposed method, we will add a discussion in the Appendix:
>
> > **Theorem 1** formally establishes that pseudo-labels produced by a model trained in a *label subspace*—one that deliberately excludes specific disturbing labels—are provably closer, in Bayes-optimal sense, to the ground-truth than labels generated in the full space. This result reframes the denoising challenge from “identifying correct labels within a noisy full space” to the more tractable task of “reducing the label space to eliminate interference.” It directly motivates RPLG’s **multi-branch auxiliary model**, where each branch deliberately trains in a subspace that omits a single candidate disturbing label.
>
> > **Theorem 1** also reveals that **no single subspace** can neutralize every possible type of label disturbance, because different instances are affected by different distractors. Consequently, an ensemble of subspace-trained branches is necessary to cover the full spectrum of disturbances. RPLG’s **meta-learned weight vector** in Eq. (11) operationalizes this insight: for every instance $x_i$, the weights $w_i$ are learned on-the-fly to emphasize branches whose excluded label is indeed the current instance’s primary distractor, thereby overcoming the limitation of single-subspace models.
>
> > **Theorem 2** further quantifies this consistency under the multi-class Tsybakov condition [29, 48], providing a lower bound on the probability that the subspace-trained model’s pseudo-labels align with the Bayes optimal classifier. This bound guarantees that, under mild statistical assumptions, the method’s pseudo-labels are reliably close to the optimal solution.
>
> ---
>
> **W2**: Novelty compared to prior multi-branch works
>
> **E2**: Thank you for helping us further enhance novelty. Although multi-branch architectures have been explored, our key contribution is the **principled exclusion of disturbing labels** within each branch—an approach grounded in a formal theoretical objective, rather than a mere engineering heuristic. Prior multi-branch studies such as [1,2,3] primarily pursue ensemble robustness and neither leverage label-subspace theory nor adopt posterior-shift reasoning. We will explicitly highlight this distinction in the Related Work section. [1] Lan et al. Knowledge Distillation by On-the-Fly Native Ensemble. NIPS. [2] Chen et al. Semi-supervised Learning with Multi-Head Co-Training. AAAI. [3] Wu et al. Multi-head mixture-of-experts. NIPS.
>
> ---
>
> **W3**: More detailed study on α
>
> **E3**: Figure 1 (Page 8) provides a sensitivity analysis of $ \alpha $ on CIFAR-10, but we agree that a more detailed investigation would be valuable. Accordingly, we have additionally conducted sensitivity analyses of $ \alpha $ on CIFAR-100 and TinyImageNet, with the results presented in Table 1. As shown, $ \alpha $ values around 0.3 yield superior performance for RPLG, and its performance remains relatively stable within the interval [0.1, 0.7]—this demonstrates the robustness of RPLG.
>
> **Table 1.** _Classification accuracy of RPLG when $\alpha$ varies on CIFAR-100 and TinyImageNet._
>
> |$\alpha$|0.1|0.3|0.5|0.7|0.9|
> |--------|----|----|----|----|----|
> |CIFAR-100|63.21|**65.03**|64.45|64.03|60.31|
> |TinyImageNet|36.97|**40.74**|39.34|38.95|33.34|
>
>
> ---
>
> **W4/L1**: Comparison of meta-weighting to alternative fusion methods
>
> **E4**: We appreciate your further assistance in helping us highlight the advantages of introducing the meta-learner. As suggested, we formulate three variants of RPLG:
>
> > (1) RPLG-S (self-training): The weight of each branch for sample $x_i$ is determined by the confidence of the branch's output on $x_i$ in the previous epoch, i.e., $w_i^j=\frac{\varphi^{j}(z_i;\Omega_j^{(t-1)})}{\sum_{k=1}^c\varphi^{k}(z_i;\Omega_k^{(t-1)})}$, where $\varphi^{j}(z_i;\Omega_j^{(t-1)})$ is the output of the $j$-th branch at epoch $t-1$.
>
> > (2) RPLG-A (attention): A lightweight attention module (2-layer fully connected network + Softmax) is introduced to generate weights based on sample features $z_i$, i.e., $w_i=\text{Attention}(z_i;\Theta_{\text{att}})$, where $\Theta_{\text{att}}$ is optimized jointly with the main model and auxiliary model.
>
> > (3) RPLG-E (ensemble-based reweighting): Weights are assigned based on the overall performance of branches on the validation set. Here, $ w^j = \frac{Acc_{j}}{\sum_{k=1}^{c} Acc_{k}}$, where $ Acc_{j} $ is the validation accuracy of the j-th branch, and the same weights are shared across all samples.
>
> Table 2 presents the performance of RPLG and its variants with alternative fusion, which demonstrates that the meta-learning-based weight strategy in RPLG has advantages over alternative weight learning or fusion methods in adapting to instance-dependent partial label scenarios, further substantiating the superiority of our meta-learner.
>
> **Table 2.** _Classification accuracy of RPLG and variants with alternative fusion on CIFAR-100 and TinyImageNet._
>
> |Method|CIFAR-100|TinyImageNet|
> |------|---------|------------|
> |RPLG|**65.03**|**40.74**|
> |RPLG-S|61.34|36.43|
> |RPLG-A|61.21|36.28|
> |RPLG-E|63.28|38.25|
>
>
> ---
>
> **Q1**: Evaluation on Uniform-PLL
>
> **E5**: As suggested, we conducted an experiment on uniform PLL settings follow [1,2] (uniformly sampling a candidate label set from the whole label space). Table 3 presents the performance of RPLG and uniform PLL baselines on KMNIST, FMNIST and CIFAR-10 in Table 3. We could observe that RPLG achieves superior or at least comparable performance to other approaches, confirming broader applicability of our approach. [1] Feng et al. Provably Consistent Partial-Label Learning. NIPS. [2] Zhang et al. Exploit Class Activation Value for Partial-Label Learning. ICLR.
>
> **Table 3.** _Classification accuracy of RPLG and compared methods on KMNIST, FMNIST and CIFAR-10 for uniform PLL._
>
> |Dataset(Backbone)|Method|Accuracy|
> |-|-|-|
> |KMNIST(LeNet)|CC|93.83±0.20%|
> ||RC|94.01±0.15%|
> ||LW|91.52±0.65%|
> ||PRODEN|93.94±0.18%|
> ||CAVL|**93.25±0.21%**|
> ||RPLG|93.12±0.10%|
> |FMNIST(LeNet)|CC|88.96±0.14%|
> ||RC|89.51±0.11%|
> ||LW|88.28±0.33%|
> ||PRODEN|89.23±0.12%|
> ||CAVL|89.99±0.10%|
> ||RPLG|**90.10±0.19%**|
> |CIFAR-10(ResNet)|CC|76.78±0.33%|
> ||RC|78.56±0.37%|
> ||LW|78.08±0.66%|
> ||PRODEN|78.72±0.48%|
> ||CAVL|79.10±0.25%|
> ||RPLG|**79.38±0.19%**|

---

> > ### Comment · Reviewer_LSa6 · 2025-08-04
> >
> > Thanks for the responses to my concerns. I will keep positive to this paper.

---

### Official Review · Reviewer_GUqm · 2025-07-02

**Clarity:** 2
**Significance:** 3
**Originality:** 3
**Rating:** 4
**Confidence:** 4

**Summary:**

This paper addresses a key challenge in Instance-dependent Partial Label Learning (ID-PLL). The authors identify that existing methods, which rely on the model's self-identification capability, often encounter a training bottleneck because they struggle to distinguish the true label from incorrect candidate labels that are strongly correlated with instance features. To overcome this, the paper proposes a novel method, RPLG (Reduction-based Pseudo-label Generation). The core idea is to train the predictive model using "reduction-based pseudo-labels". These are generated by a multi-branch auxiliary model where each branch is trained in a "label subspace" that explicitly excludes one specific label, thereby avoiding its interference. The framework further incorporates a meta-learner to perform a weighted aggregation of the outputs from all branches to produce the final pseudo-label. Theoretically, the authors prove that their proposed pseudo-labels exhibit greater consistency with the Bayes optimal classifier.

**Questions:**

1. Must the number of branches equal the number of labels? Have you considered using fewer branches, such as via label clustering or dimensionality reduction, to reduce computation?
2. How does the method handle labels that are both correct and disturbing in different samples?

**Ethical Concerns:**

["NO or VERY MINOR ethics concerns only"]

**Final Justification:**

The authors have successfully adressed the majority of my concerns as mentioned in the rebuttal.

**Limitations:**

1. The method trains one branch per label. As the number of classes increases (e.g., 200 in TinyImageNet), the computation and memory cost grows rapidly.
2. The theoretical guarantees rely on ideal assumptions, such as the model closely approximating the posterior distribution and satisfying the Tsybakov condition.

**Quality:**

3

**Strengths And Weaknesses:**

Strengths:
1. The concept of "reduction-based pseudo-labels" and the use of label subspaces to mitigate interference  is a highly novel contribution. The approach directly targets a well-articulated shortcoming in prior work, namely the model's inability to handle "disturbing incorrect labels", rather than offering a mere incremental improvement.
2. The method is not just heuristic but is supported by a solid theoretical analysis. The authors provide theorems (Theorem 1 and 2) to formally justify the approach, demonstrating that the generated pseudo-labels are more consistent with the Bayes optimal classifier compared to those from the predictive model itself.
3.The paper presents comprehensive and convincing experimental results. RPLG is shown to significantly outperform a wide range of baselines, including recent state-of-the-art methods, across various benchmark and real-world datasets. The ablation study comparing RPLG with RPLG-NM effectively validates the contribution of the meta-learning component.

Weaknesses:
1. The multi-branch model design incurs significant overhead when the number of classes is large. For example, TinyImageNet training time is substantially higher.
2. While the paper uses a weight vector to fuse predictions from each branch, it does not analyze how the branches interact or whether some are redundant.
3. While a variant without meta-learning (RPLG-NM) is compared, the necessity of using one branch per label is not ablated or justified. Could fewer branches suffice?
4. All baselines use the same backbone and data augmentation, but many of them are not designed for ID-PLL. It would strengthen the case to compare against ensemble-based or meta-learning based approaches beyond the PLL field.

---

> ### Author Rebuttal · Authors · 2025-07-31
>
> We sincerely appreciate your meticulous review and professional feedback on our paper. Your recognition of **the novelty of the "reduction-based pseudo-labels" concept, the use of label subspaces to mitigate interference, and the solid theoretical analysis** is highly encouraging. In response to Weaknesses (W), Questions (Q) and Limitations (L), we would like to provide the following Explanations (E).
>
> ---
>
> **W1/L1**: Linear growth of branches with classes leading to high computational overhead
>
> **E1**:  As highlighted in Section 4.4 and Appendix A.4, our framework mitigates this through two key designs: (1) **shared feature extractors with parallel training of branches**, which reduces redundant parameter storage and computation; (2) **online meta-learning optimization**, which avoids exhaustive bi-level optimization. Experimental results in Table 4 show that RPLG incurs only a marginal linear increase in training time (e.g., 6.38 hours on TinyImageNet vs. 5.84 hours for the second-ranked algorithm baseline POP), which is manageable given the significant performance gains (40.74% vs. 39.27% accuracy on TinyImageNet).
>
> Additionally, as noted earlier, we formulate variants of RPLG with fewer branches, considering the trade-off between performance and computational cost, which will be elaborated on in the subsequent section **E3**.
>
> ---
>
> **W2**: Branch interaction and redundancy
>
> **E2**: Branch interaction is implicitly realized through a reduction-based pseudo-label in an iterative process: the outputs of all branches for instance $ x_i $ are integrated into the reduction-based pseudo-label $ v_i $ via meta-learned weights $ w_i $ in Eq. (8), and in the next epoch, the pseudo-label is normalized into $ U_i $ in Eq. (9) for training the branches.
>
> Branch redundancy is addressed by the **meta-learned weight vector** in Eq. (11-17). The meta-learner $ g(\cdot; \Gamma) $ adaptively assigns weights to branches based on instance features, effectively suppressing redundant branches. For example, if a branch trained by excluding label $ j $ is irrelevant to instance $ x_i $ (i.e., $ j $ is not a disturbing label for $ x_i $), its weight $ w_i^j $ becomes negligible. This is validated by Table 3 in the submission, where RPLG (with meta-weights) outperforms RPLG-NM (uniform weights) across all datasets, demonstrating that learned weights eliminate redundancy.
>
> ---
>
> **W3/Q1**: Necessity of one-branch-per-label
>
> **E3**: This is an insightful point. One-branch-per-label is a universal and conservative scheme to deal with Instance-Dependent Partial Label Learning (ID-PLL), since each class may act as a disturbing class for other classes.
>
> In practice, when efficiency is preferred, using fewer branches via label clustering or dimensionality reduction is a good idea. Here, we formulate two variants of RPLG with fewer branches: RPLG-W and RPLG-R. RPLG-W employs Word2Vec for label clustering to group labels into a cluster with similar textual semantics (e.g., "dog" and "cat"). It retains only those branches where the excluded label belongs to the cluster—this is because labels with similar semantics are more likely to interfere with each other, so training a series of subspaces that mutually exclude semantically similar labels is helpful. In contrast, RPLG-R retains branches randomly. Table 1 presents the performance of these two variants on TinyImageNet. From Table 1, we can observe that compared with randomly retaining branches, preserving branches that exclude semantically similar labels achieves, to some extent, a favorable trade-off between performance and computational overhead.
>
> **Table 1.** _Classification accuracy and time consuming of our approach RPLG and variants with fewer branches on TinyImageNet._
>
> |Method|Retaining Rate|Average Accuracy(%)|Time(h)|
> |---|---|---|---|
> |RPLG|100%|40.74|6.38|
> |RPLG-W|80%|40.63|6.30|
> ||60%|40.27|6.23|
> ||40%|39.89|6.09|
> ||20%|39.53|5.97|
> |RPLG-R|80%|39.65|6.30|
> ||60%|39.05|6.23|
> ||40%|38.78|6.09|
> ||20%|38.84|5.97|
> |POP (ranked second)| - |39.27|5.84|
>
>
> ---
>
> **W4**: Comparison against ensemble-based or meta-learning-based approaches beyond the PLL field
>
> **E4**: We have adapted MeanTeacher (a classic ensemble-based method for semi-supervised learning) [1] and Meta-Weight-Net (a classic meta-learning-based method for noisy label learning) [2]into versions for PLL learning by replacing the corresponding loss with the representative Proden Loss [3]. Table 2 presents the performance comparison between these two methods and our algorithm on CIFAR-100 and TinyImageNet. [1] Antti Tarvainen et al. Mean teachers are better role models: Weight-averaged consistency targets improve semi-supervised deep learning results. NIPS. [2] Shu et al. Meta-Weight-Net: Learning an Explicit Mapping For Sample Weighting. NIPS. [3] Lv et al. Progressive Identification of True Labels for Partial-Label Learning.ICML
>
> **Table 2.** _Classification accuracy of RPLG and methods beyond the PLL field on CIFAR-100 and TinyImageNet._
>
> |Method|CIFAR-100|TinyImageNet|
> |------|---------|------------|
> |RPLG|**65.03**|**40.74**|
> |MeanTeacher|62.73|34.86|
> |MetaWeightNet|62.64|33.91|
>
> ---
>
> **Q2**: Handling labels that are both correct and disturbing across different samples
>
> **E5**: Our approach RPLG addresses this through two core designs: multiple branches for label subspace training and a meta-learner for aggregation weights. For instance, if label $ j $ is disturbing for $ x_1 $ but correct for $ x_2 $, the meta-learner will learn to assign a larger weight $ w_1^j $ to branch $ j $ (trained in the label subspace excluding label $j$) during aggregation—this avoids the interference of label $ j $ on $ x_1 $. Conversely, for $ x_2 $, it will learn to assign a smaller $ w_2^j $ while increasing weights $ w_2^{k \neq j} $, thereby preserving the signal from label $ j $.
>
> ---
>
> **L2**: Assumptions in the theorems
>
> **E6**: In our work, all assumptions are either restrictive or empirically grounded, and thus are acceptable considering that it is necessary to ensure tractable analysis. For instance, while we assume the model can approximate the posterior distribution, the extent of this approximation is controlled by $\epsilon$, which is also an extension or relaxation of the assumption used in the earlier PLL work [1]. Additionally, the Tsybakov condition, which pertains to the dataset distribution, has been experimentally validated in the prior study [2]. [1] Feng et al. Provably Consistent Partial-Label Learning. NIPS. [2] Zheng et al. Error-Bounded Correction of Noisy Labels. ICML.

---

### Note · Authors · 2025-08-12

We sincerely thank the Area Chairs and Reviewers for their meticulous evaluation and constructive feedback, which have significantly improved our work. Below, we summarize the core contributions, key responses to reviewers’ concerns, and the scholarly value of our submission, to provide a concise perspective for the Area Chairs’ deliberation.

### 1. Novel Contributions to Instance-Dependent Partial Label Learning

We propose **Reduction-based Pseudo-label Generation (RPLG)**, a theoretically grounded framework that addresses a central challenge in Instance-dependent Partial Label Learning (ID-PLL): disentangling true labels from feature-correlated incorrect labels. RPLG introduces **label subspace training**, where each branch of a multi-branch auxiliary model is trained within a subspace that excludes one specific label. Rooted in label-subspace theory and posterior-shift reasoning, this design effectively isolates disturbing labels, leading to more accurate pseudo-label generation.

### 2. Rigorous Theory and Empirical Validation

Our theoretical analysis demonstrates that reduction-based pseudo-labels exhibit stronger alignment with the Bayes optimal classifier than those generated in the full label space.

* **Theorem 1** quantifies the reduction in posterior gap in label subspaces.
* **Theorem 2** establishes a lower bound on the alignment probability under the multi-class Tsybakov condition.

Experiments on CIFAR-100, TinyImageNet, and Yahoo!News confirm consistent improvements over state-of-the-art baselines, including +1.47% on TinyImageNet and +0.95% on Soccer Player. Ablation studies further show that meta-learned weights play a critical role in avoiding branch redundancy.

### 3. Addressing Reviewers’ Concerns

* **Efficiency**: Shared feature extractors and online meta-optimization keep overhead low. The RPLG-W variant reduces branches by 20% with negligible loss.
* **Meta-learner effectiveness**: Outperforms self-training, attention, and ensemble reweighting on CIFAR-100.
* **Assumptions**: Theoretical assumptions are empirically validated or consistent with established PLL literature.

**Conclusion**

RPLG advances ID-PLL through novel theoretical insights, strong empirical evidence, and clear practical benefits. We believe these qualities meet the NeurIPS bar in both originality and impact, and we hope this perspective assists in the Area Chairs’ final decision.

---

### Decision · Program_Chairs · 2025-09-17

**Decision:**

Accept (poster)

**Comment:**

Following the final round of reviews, the paper received one "Accept" rating and three "Borderline Accept" ratings. The reviewers recognized the technical novelty of the proposed approach and acknowledged the strength of both the theoretical analysis and the empirical results. Additionally, the rebuttal played a constructive role in the evaluation process, as it effectively addressed most of the reviewers' concerns and clarified several key points raised during the initial review phase.